# IMPLICITQA: GOING BEYOND FRAMES TOWARDS IMPLICIT VIDEO REASONING

## ABSTRACT

Video Question Answering (VideoQA) has made significant strides by leveraging multimodal learning to align visual and textual modalities. However, current benchmarks overwhelmingly focus on questions answerable through explicit visual content - actions, objects, and events directly observable within individual frames or short clips. In contrast, creative and cinematic videos - such as movies, TV shows, and narrative-driven content - employ storytelling techniques that deliberately omit certain depictions, requiring viewers to infer motives, relationships across discontinuous frames with disjoint visual contexts. Humans naturally excel at such implicit reasoning, seamlessly integrating information across time and context to construct coherent narratives. Yet current benchmarks fail to capture this essential dimension of human-like understanding. To bridge this gap, we present ImplicitQA, a novel benchmark specifically designed to test VideoQA models on human-like implicit reasoning. ImplicitQA comprises $1K$ meticulously annotated QA pairs drawn from $1K$ high-quality creative video clips covering 15 genres across 7 decades of content. Questions are systematically categorized into nine key reasoning dimensions: lateral and vertical spatial reasoning, depth and proximity, viewpoint and visibility, motion and trajectory, causal and motivational reasoning, social interactions, physical context, and inferred counting. These annotations are deliberately challenging, crafted by authors, validated through multiple annotators, and benchmarked against human performance to ensure high quality. Our extensive evaluations on 11 leading VideoQA models reveals consistent and significant performance degradation, underscoring their reliance on surface-level visual cues and highlighting the difficulty of implicit reasoning. Even the best model substantially underperforms human baselines with only 64% accuracy, and no open-source model exceeds 50% accuracy. Performance variations across models further illustrate the complexity and diversity of the challenges presented by ImplicitQA. Our analysis highlights the unique challenges of implicit reasoning, including limited gains from scaling frames or parameters. By releasing both the dataset and data collection framework, ImplicitQA establishes a rigorous, diverse, and reproducible testbed for advancing VideoQA.

## 1 INTRODUCTION

Video Question Answering (VideoQA) sits at the intersection of computer vision and natural language processing, aiming to answer natural language questions based on visual content in videos. Recent progress in VideoQA Ren et al. (2023); Zhang et al. (2024a); Maaz et al. (2024); Wang et al. (2024b); Li et al. (2024a;b); Zhang et al. (2024b); Wang et al. (2024a); Bai et al. (2025); Yuan et al. (2025) has been fueled by multimodal learning techniques that integrate visual and textual modalities, enabling impressive performance on datasets where questions are grounded in explicit visual content. These benchmarks Xu et al. (2016); Yu et al. (2019); Xiao et al. (2021); Lei et al. (2018); Li et al. (2023); Cai et al. (2024); Fu et al. (2024); Rawal et al. (2024); Liu et al. (2024) typically emphasize recognizing objects, identifying actions, and understanding events that are directly observable within individual frames or short clips. However, cinematic and narrative-driven videos - such as movies frequently employ storytelling devices that challenge this paradigm. Rather than explicitly showing every key detail, such videos rely on indirect cues, subtle scene transitions, and off-screen implications to advance the plot or convey meaning. For example, a character's motive might be implied through a prior conversation but never visually depicted, or a causal event might occur off-screen, requiring

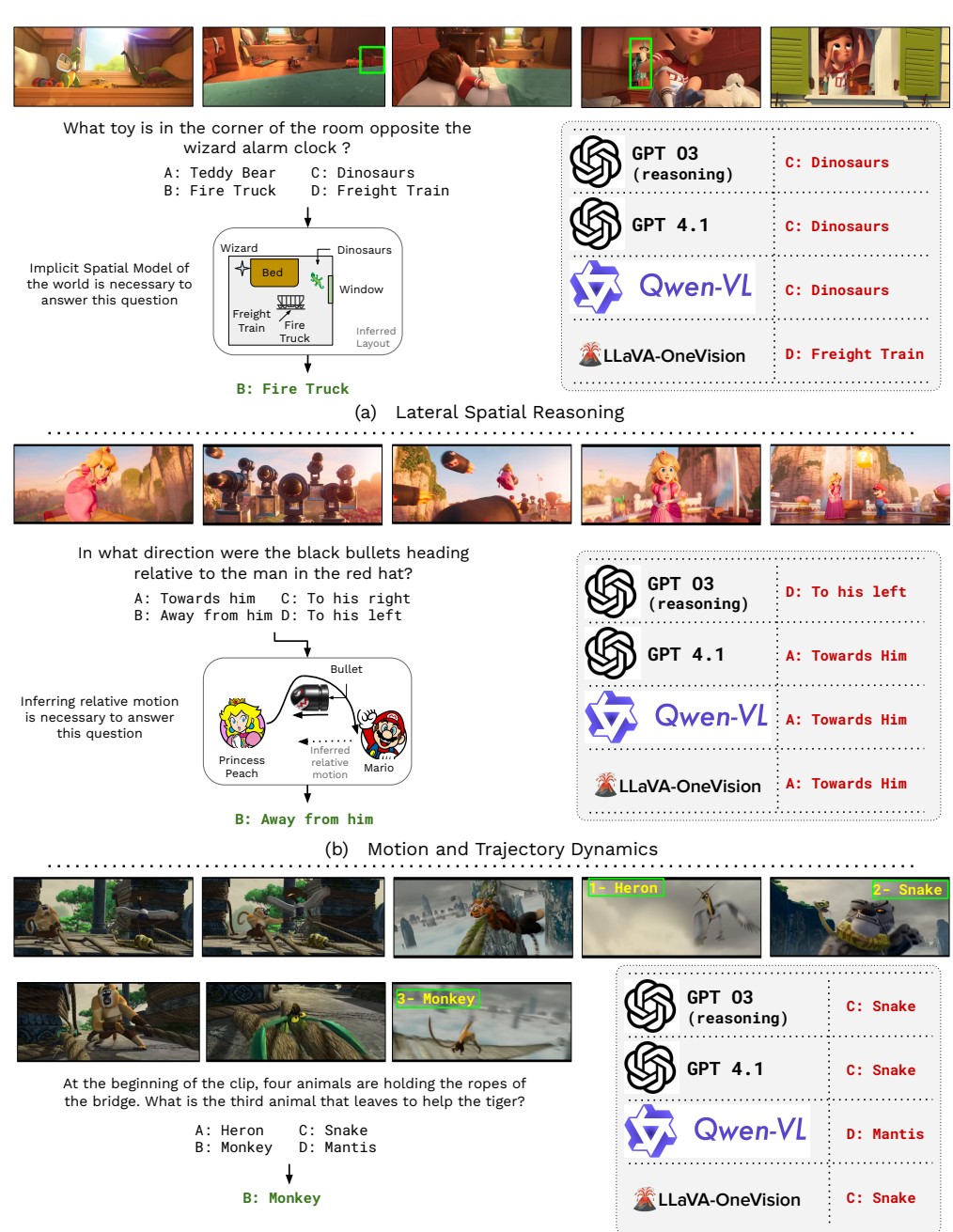

Figure 1: ImplicitQA examples, each targeting a distinct implicit-reasoning dimension. (a) **Lateral spatial reasoning**-identifying the toy opposite the wizard clock by mentally mapping objects across the scene. (b) **Motion and trajectory dynamics**-inferring that black bullets move away from Mario by integrating actions and character positions. (c) **Inferred counting**-determining which animal is the third to leave a bridge by tracking sequential departures that are never fully visible onscreen. Models that excel at explicit perception often fail on these tasks, highlighting the need for benchmarks that probe deeper narrative understanding.

viewers to infer its effects from subsequent scenes. Humans are remarkably skilled at making such implicit inferences, seamlessly connecting context, prior knowledge, and temporal cues to reconstruct missing narrative elements. This ability forms a cornerstone of real-world video understanding. However, current VideoQA benchmarks largely overlook this aspect of human-like reasoning. They reward models that excel at surface-level perception but do not challenge systems to infer beyond what is visually present. As a result, today's state-of-the-art models may perform well on existing datasets

Table 1: Comparison of `ImplicitQA` with existing VideoQA datasets. `ImplicitQA` uniquely focuses on implicit reasoning with visual content, annotated end-to-end by domain experts.

| Dataset | Tested-Abilities | | Vision | Video Source | Annotations | | Question |
| | Direct | Implicit | Only ? | | Humans | Automated | Filtering |
| --- | --- | --- | --- | --- | --- | --- | --- |
| **Cinepile** Rawal et al. (2024) | ✓ | ✗ | ✗ | Movie Clips | ✗ | QA (Templates + LLM) | LLM |
| **TVQA** Lei et al. (2018) | ✓ | ✗ | ✗ | TV Show Clips | QA (Templates) | - | - |
| **VideoMME** Fu et al. (2024) | ✓ | ✗ | ✗ | Diverse | Experts | - | LLM |
| **MVBench** Li et al. (2023) | ✓ | ✗ | ✓ | Prior Datasets | Prior Datasets | Wrong Options | - |
| **TemporalBench** Cai et al. (2024) | ✓ | ✗ | ✓ | Prior Datasets | Captions | Pairing | LLM |
| **TempCompass** Liu et al. (2024) | ✓ | ✗ | ✓ | Stock + Transforms | Class Labels | QA (Templates + LLM) | - |
| **Open-EQA** Majumdar et al. (2024) | ✓ | ✗ | ✓ | Indoor Egocentric tours | Human | - | - |
| **VSI-Bench** Yang et al. (2025) | ✓ | ⚠️ | ✓ | Indoor Egocentric tours | QA (Templates) | QA (Templates) | Human |
| **ImplicitQA** | ✗ | ✓ | ✓ | Movie Clips | Experts (Video selection + QA + Filtering) | | |

while failing to grasp deeper, implicit reasoning. Consider the example depicted in Figure 1(b), where the question posed is: "In what direction were the black bullets heading relative to the man in the red hat?" with answer choices: {'A': 'Towards him', 'B': 'Away from him', 'C': 'To his right', 'D': 'To his left'}. The correct answer is "B" (Away from him), as the scene implicitly shows the princess Peach running towards Mario while black bullets are fired towards her, thus moving away from Mario. However, existing VideoQA models consistently answer incorrectly, choosing "Towards him," failing to grasp the information across frames. Even the O3 model incorrectly selects "To his left," highlighting the widespread difficulty in capturing implicit narrative dynamics.

To address this critical gap, we introduce `ImplicitQA`, a benchmark designed to probe the limits of implicit reasoning in VideoQA. We collect $1K$ carefully curated question-answer pairs from $1K$ diverse video clips sourced from movies. Unlike traditional benchmarks, `ImplicitQA` focuses exclusively on questions that cannot be answered through direct observation of frames alone. Instead, they require reasoning about unstated character motives, social interactions, physical context, and other narrative nuances that are implied but not explicitly depicted.

Our dataset is organized into 9 core reasoning categories: (1) Lateral Spatial Reasoning, (2) Vertical Spatial Reasoning, (3) Relative Depth and Proximity, (4) Motion and Trajectory Dynamics, (5) Viewpoint and Visibility, (6) Motivational Reasoning, (7) Social Interaction and Relationships, (8) Physical and Environmental Context, and (9) Inferred Counting. We have annotated the data ourselves, thus ensuring that the questions are both challenging and aligned with the nuanced reasoning capabilities we aim to benchmark. Further, we re-verify the annotations amongst ourselves, this curation process also minimizes the ambiguity and guarantees that each question tests meaningful aspects of implicit understanding.

We evaluate on prominent VideoQA models on `ImplicitQA` and observe significant performance drops compared to standard benchmarks. This finding underscores the current limitations of VideoQA systems, which remain heavily reliant on explicit visual cues. Notably, we find that reasoning-oriented models outperform non-reasoning models: for example, GPT-o3 OpenAI (2024b) achieves a 9.8% higher accuracy than GPT-4.1 OpenAI (2024a). This gap illustrates the necessity of deeper reasoning capabilities to tackle the challenges posed by `ImplicitQA`, further validating our focus.

In summary, `ImplicitQA` raises a new research challenge: to build models capable of deep temporal reasoning and implicit inference across frames - moving VideoQA closer to true human-like video understanding. Our contributions are listed below:

- We introduce `ImplicitQA`, the first benchmark designed to test implicit reasoning in VideoQA, focusing on questions that require inference beyond direct visual observations.
- We manually curate a high-quality dataset of 1k QA pairs across 1k diverse video clips, with annotation conducted by experts in computer vision to ensure rigor and relevance.
- We define a taxonomy of 9 categories, covering lateral spatial reasoning, depth and proximity, social dynamics, and more, to facilitate targeted analysis and benchmarking.
- We benchmark SoTA VideoLLMs on `ImplicitQA` and reveal significant performance degradation, highlighting the gap between current capabilities and true narrative understanding.

## 2 RELATED BENCHMARKS

We compare our proposed dataset against some recent VideoQA benchmarks, which we broadly categorize into Vision-only benchmarks, and Vision and speech benchmarks, which require integration of information from othe modalities like speech.

## 2.1 VISION ONLY BENCHMARKS

**MVBench** Li et al. (2023): MVBench aggregates roughly 4,000 human- or automatically–derived multiple-choice questions drawn from 11 public age datasets, pairing each 5–35 s clip with four or five answer options. A "static-to-dynamic" pipeline converts image-based tasks into temporally grounded ones, ensuring coverage of both short-term motions and longer-horizon causal phenomena.

**TempCompass** Liu et al. (2024): TempCompass focuses on pure temporal manipulation by algorithmically editing 410 royalty-free clips ($\leq$30 s) into pairs/triplets that differ only in one temporal property (e.g., playback reversal, reordered events, speed changes). GPT-3.5 then generates 7,540 diverse tasks (multiple-choice, yes/no, caption matching, constrained captioning).

**TemporalBench** Cai et al. (2024): TemporalBench curates 2,032 silent clips (<20 s) from seven public corpora, enriches each with a dense human-written caption plus up to 15 machine-crafted counterfactual captions, and forms 9,867 contrastive pairs targeting fine-grained temporal distinctions (action order, frequency, direction, effector).

**Open-EQA** Majumdar et al. (2024) and **VSIBench** Yang et al. (2025): Both of these datasets are based one pre-existing 3d scans of indoor environments taken from an egocentric POV. As a result the videos in these datasets carry a dense view of the environment, which needs to be stitched together to answer questions. Due to the dense nature of the sampling, these videos primarily require limited implicit spatial reasoning (indicated by ⚠ symbol). VSI-Bench spans configuration, measurement, and spatiotemporal tasks (e.g., room size, appearance order); OpenEQA covers seven categories (object/attribute/state recognition, localization, spatial and functional reasoning, world knowledge).

ImplicitQA is designed to go beyond existing video benchmarks: Firstly, Implicit, multi-frame inference. Questions demand reasoning about off-screen events, unstated motives, or causal chains spanning multiple clips, no single-frame cues suffice. Secondly, we utilize expert manual curation. Unlike MVBench's mixed annotations, TempCompass's algorithmic edits, or TemporalBench's counterfactual pipeline, our 1K questions from 1k clips are manually authored and rigorously cross-verified by computer-vision experts, ensuring each item truly probes implicit visual reasoning.

## 2.2 VISION AND SPEECH FUSION BENCHMARKS

**VideoMME** Fu et al. (2024) VideoMME comprises 900 expert-annotated YouTube videos (short <2 min, medium 4–15 min, long 30–60 min) with frames, raw audio, and automatically extracted subtitles. Human annotators author balanced multiple-choice questions across diverse domains, and a "text-only" filter removes any item solvable without visual or acoustic cues.

**TVQA** Lei et al. (2018) TVQA provides $\sim$ 22 K clips (60–90 s) from six U.S. TV series, each paired with dialogue subtitles and precise timestamps. Its compositional "WH-word ... when/before/after ..." templates force joint vision–language reasoning and moment localization, but the narrow domain and reliance on spoken dialogue limit purely visual inference.

**CinePile** Rawal et al. (2024) CinePile curates $\sim$ 9K movie snippets ($\sim$ 160 s) from MovieClips, augmented with professional audio descriptions and subtitles. An LLM-driven pipeline yields 304 K multiple-choice QAs, filtered adversarially for shortcut resistance.

ImplicitQA departs from these benchmarks in three key ways: Firstly, implicit multi-frame inference: every question demands reasoning about off-screen events, unstated motives, or causal chains not solvable by a single frame or subtitle snippet. Secondly, Visual-only focus: We strip away subtitles and audio tracks entirely, models must extract and integrate visual cues across frames without any textual crutch. Finally, expert manual curation: Unlike datasets that rely on automated pipelines or narrow scripted dialogue, our 1K questions across 1K clips are *manually authored and cross-verified* by computer-vision experts, ensuring each item truly probes implicit visual reasoning.

## 3 DATASET CURATION

A critical component of constructing the `ImplicitQA` benchmark was the creation of high-quality, challenging question-answer pairs that test implicit reasoning over video content. To this end, we developed a custom annotation tool specifically designed to streamline and standardize the data collection process for implicit reasoning in videos.

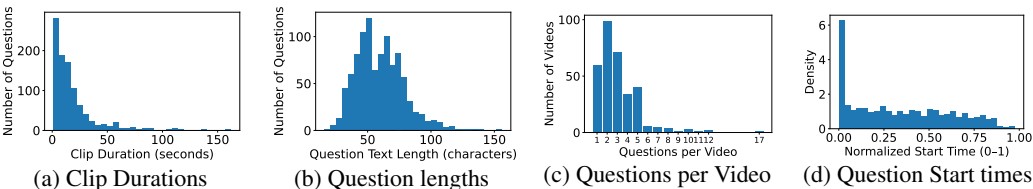

Figure 2: ImplicitQA Curation Pipeline. We begin by selecting creative video clips and download them. An expert-annotator pool then uses our FrameQuiz Annotation Tool to (1) mark temporal segments, (2) add a multiple-choice question and its correct answer for the segment, and (3) craft plausible distractor options. These annotated clips form the raw ImplicitQA Dataset. Next, a non-expert annotator pool employs the ImplicitEval Annotation Tool to answer each question, yielding a human baseline accuracy score. We run GPT-4.1OpenAI (2024a) on the dataset to automatically assign initial category tags, which are then relabeled by the expert annotators.

| (a) Clip Durations | (b) Question lengths | (c) Questions per Video | (d) Question Start times |

Figure 3: Visualization of ImplicitQA statistics.

## 3.1 ANNOTATION TOOL AND DATA COLLECTION

Our custom-built annotation interface was designed to provide an intuitive and efficient workflow. The tool allowed annotators to: Watch a video clip directly within the interface, select start and end timestamps marking the temporal window relevant to the annotated question. Write the question and corresponding answer choices, explicitly tying them to the selected video segment. This structured approach ensured that each question was clearly linked to a specific portion of the video, even if the reasoning required drawing upon broader context across multiple scenes. The tool was optimized for fast navigation, enabling annotators to pause, rewind, or step through clips frame-by-frame to closely examine nuanced visual cues. Please refer to Supplementary Section 4 for interface figure.

To ensure annotation quality and reliability, we implemented a save-and-replay feature within the tool, enabling annotators to revisit their annotated segments, replay the selected video portion, and iteratively refine or validate their annotations before final submission. We also verify the annotations amongst ourselves. This process ensured that questions accurately aligned with the video context and targeted implicit rather than explicit reasoning. We ourselves have annotated questions with the intent to probe deeper inferential reasoning rather than relying solely on directly observable content. Thus the author annotation process contributed to both the conceptual depth and technical relevance of the dataset. For video selection, we curated a diverse set of 1K creative videos, comprising movies of varied genres and mediums (3D animated and live-action) known for employing narrative techniques such as implied causality, off-screen action, symbolic representation, and indirect storytelling. We prioritized scenes that challenge viewers to make inferences beyond directly visible actions or objects, focusing on content where critical narrative elements are omitted, subtle, or distributed across frames. Statistical characteristics of our collected dataset can be seen in Figure 3.

**Question Diversity.** To measure question diversity quantitatively, we compute mean pairwise cosine similarity of sentence embeddings between our questions. This metric for text diversity has previously been used in the literature Tevet & Berant (2021). We utilize the all-MiniLM-L6-v2 model from the Sentence Transformers library to compute text embeddings. The results of this analysis are provided in the Table 2. Lower mean similarity between questions indicates higher diversity, as can be seen our questions have higher diversity compared to prior works. We further provide dataset diversity in terms of genre, movie release timeline, media type in Section C.

Table 2: Mean Pairwise Similarity for Question Embeddings. (lower = more diverse)

| Benchmark | MPS ($\downarrow$) |
|---|---|
| MVBench | 0.293 |
| TempCompass | 0.228 |
| Cinepile | 0.216 |
| Movie-QA | 0.191 |
| Implicit-QA | **0.161** |

## 3.2 DATASET CATEGORIZATION

To ensure comprehensive coverage of the diverse reasoning abilities required for implicit video understanding, we organized the dataset into nine distinct reasoning categories,

each targeting a specific type of implicit inference. Below, we formally define each category of implicit reasoning. We present qualitative examples in Figure 1, 6 to highlight the implicit nature of questions. We also show differences in temporal length between categories in Figure 4, with counting and vertical spatial reasoning videos being the longest.

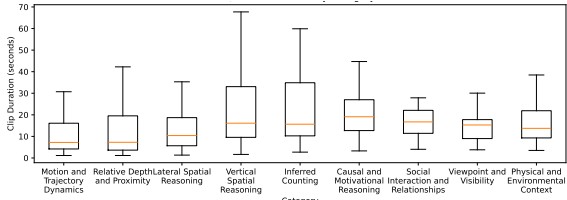

Figure 4: Question durations for each category.

**Lateral Spatial Reasoning** Questions in this category test the ability to infer spatial relationships, positions, or arrangements of objects and characters along lateral orientations. They require viewers to implicitly track or reason about relative positions without explicit directional guidance.

**Vertical Spatial Reasoning** This category assesses the viewer's capacity to implicitly reason about spatial relationships, positions, or arrangements of objects and characters along a vertical axis (above-below orientation). Questions often involve interpreting hierarchical arrangements or vertical positioning that aren't explicitly depicted.

**Relative Depth and Proximity** These evaluate the ability to infer relative distances, depth perception, and proximity between characters or objects within the scene. They require implicit judgments about which objects or characters are closer or further from the viewer or each other, without explicit depth cues.

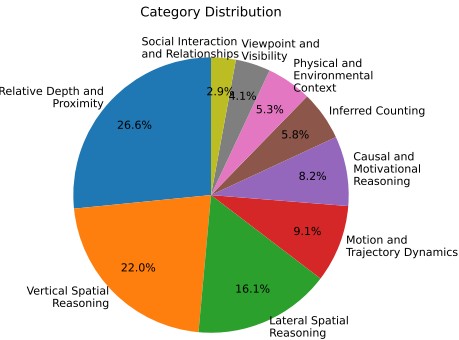

Figure 5: Distribution across categories

**Viewpoint and Visibility** Questions in this category require inferring what is observable from a particular vantage point, whether it be a character's perspective or camera angle. They must reason about line-of-sight, occlusions, and spatial orientations.

**Motion and Trajectory Dynamics** Questions in this category assess the ability to implicitly track motion, movement directions, and trajectories of characters or objects across discontinuous frames. These movement patterns might be implied and not fully observable in a single scene.

**Motivational Reasoning** Questions in this category require viewers to infer character motives, or likely future events based on incomplete or indirect visual information. These questions emphasize unstated cause-effect chains within the story.

**Inferred Counting** This category involves implicit counting or enumeration tasks that require aggregating scattered visual evidence across multiple frames or scenes. Such questions demand sustained attention and integration of visual clues over time to infer quantitative details.

**Physical and Environmental Context** Questions in this category probe reasoning about physical elements of the environment, as well as environmental dynamics that may be implied through narrative or partial visual cues but not overtly shown.

**Social Interaction and Relationships** This category captures reasoning about social dynamics, interactions, and relationships between characters that are inferred through subtle or indirect cues. These questions require understanding of unspoken social behaviors or contextual relational information.

## 4 BENCHMARKING

Our evaluation of 30 VideoQA model configurations across different scales and temporal contexts on the `ImplicitQA` benchmark reveals several critical insights. Detailed table is presented in Section B. We evaluate a broad range of open-source and proprietary multimodal models on the `ImplicitQA` benchmark, focusing on their ability to perform implicit reasoning over videos. The evaluation includes multiple model families, scales, and video context lengths. We include the

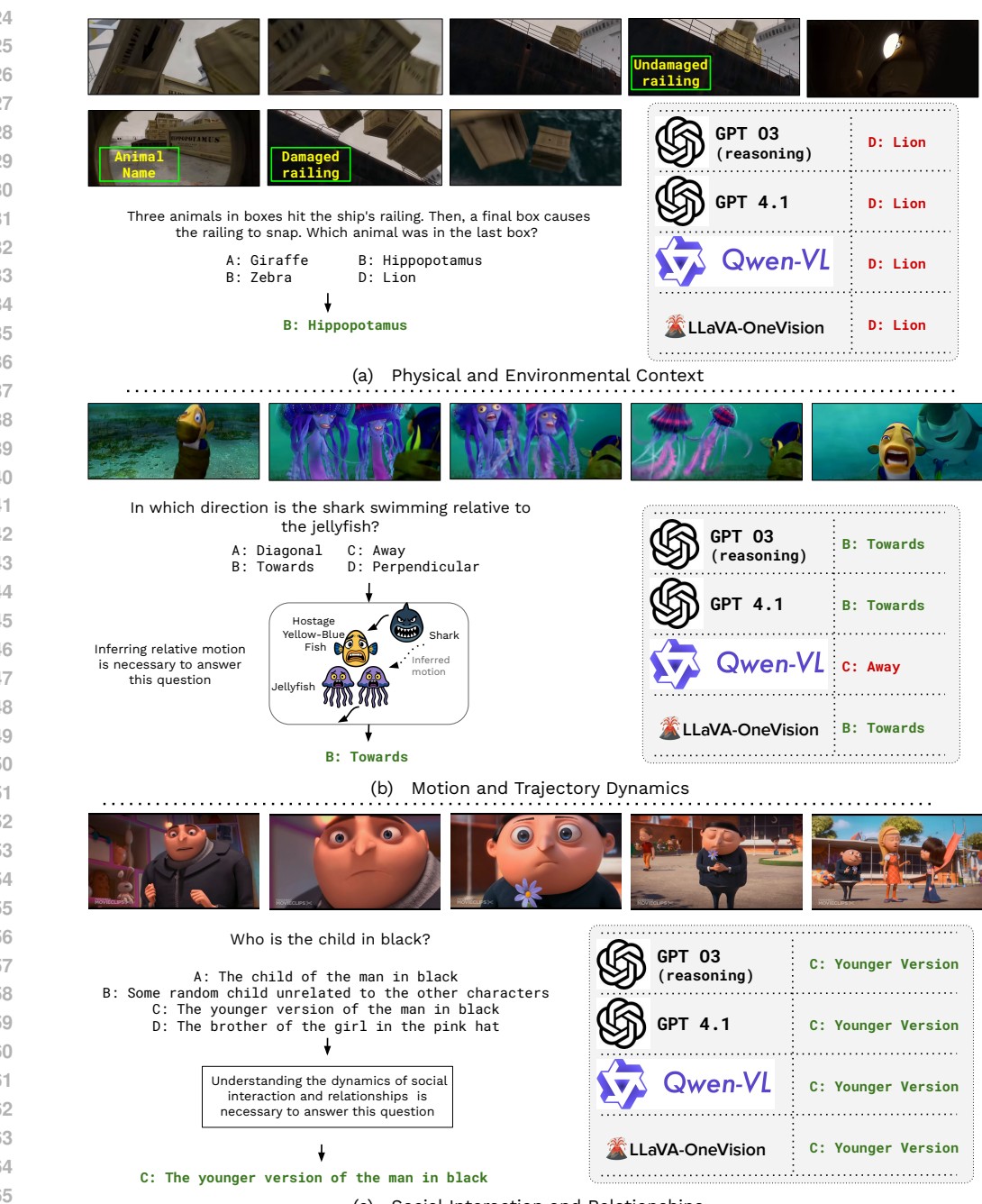

Figure 6: More Qualitative `ImplicitQA` examples, targeting distinct implicit-reasoning dimension.

following models in our evaluation: (1) Open-source models: LLaVA-NeXT-Video Zhang et al. (2024a), LLaVA-OneVision Li et al. (2024a), LLaVA-Video Zhang et al. (2024b), Qwen2 VL Wang et al. (2024a), Qwen2.5 VL Bai et al. (2025), InternVL3 Zhu et al. (2025), Gemma 3 Team et al. (2025), (2) Closed-source models: GPT-4.1 OpenAI (2024a)-full, mini, and nano variants, and the reasoning based GPT-O3 OpenAI (2024b) model, Gemini 2.5 Flash Comanici et al. (2025), Claude 4 sonnet Anthropic. We analyze the performance of these models via category-wise accuracy, and overall average accuracy. We also report the macro-average accuracy, which is the average of category-wise accuracies giving equal weight to each category. Additionally, we explore the impact of integrating an explicit reasoning prompt with GPT models. Please refer to the Section B for detailed scores at model scale and number of frames.

Table 3: Results on `ImplicitQA` with 16 input frames. **Best** and second best results are highlighted.

| Model | Lateral Spatial Reasoning | Vertical Spatial Reasoning | Relative Depth and Proximity | Viewpoint and Visibility | Motion & Traj. Dynamics | Causal & Motivational Reasoning | Inferred Counting | Physical & Env. Context | Social Interaction & Relations | Avg. | Macro Avg. |
|---|---|---|---|---|---|---|---|---|---|---|---|
| Human Baseline | 85.4 | 79.1 | 80.4 | 90.0 | 91.9 | 94.4 | 65.9 | 83.3 | 100.0 | 83.0 | 85.6 |
| **Open-Weight Models (7B-Scale)** | | | | | | | | | | | |
| LLaVA-Next-Video | 36.0 | 29.6 | 30.1 | 48.8 | 36.3 | 39.0 | 30.2 | 35.7 | 51.7 | 33.9 | 37.5 |
| LLaVA-OneVision | 37.3 | 46.8 | 35.0 | 56.1 | 57.1 | 57.3 | 23.3 | 50.0 | 55.2 | 43.4 | 46.4 |
| LLaVA-Video | 36.0 | 44.0 | 31.6 | 56.1 | **60.4** | 62.2 | 14.0 | 50.0 | 62.1 | 42.1 | 46.3 |
| Gemma 3 | **48.5** | 38.9 | 32.3 | **68.3** | 39.6 | 59.8 | 25.6 | **57.1** | 58.6 | 42.1 | 47.6 |
| InternVL 3 | 34.8 | 39.4 | 37.2 | 56.1 | 51.7 | **64.6** | **34.9** | **57.1** | **75.9** | 43.3 | **50.2** |
| Qwen2-VL | 39.8 | 46.8 | **40.6** | 51.2 | 52.7 | 58.5 | 16.3 | 35.7 | 72.4 | **44.9** | 46.0 |
| Qwen2.5-VL | 41.6 | **47.2** | 32.7 | 61.0 | 50.5 | 51.2 | 25.6 | 42.9 | 62.1 | 42.8 | 46.1 |
| **Proprietary Models** | | | | | | | | | | | |
| Gemini 2.5 Flash | 41.6 | 57.7 | 32.0 | 45 | 60.4 | 71.3 | 42.5 | 69.2 | 81.5 | 49.6 | 55.7 |
| Claude 4 Sonnet | 39.1 | 44.9 | 37.9 | 43.9 | 52.8 | 74.4 | 23.3 | 64.3 | 72.4 | 45.4 | 50.3 |
| GPT 4.1 | 42.9 | 53.2 | 51.1 | 48.8 | 59.3 | 82.9 | 41.9 | 71.4 | 75.9 | 54.3 | 58.6 |
| GPT O3 | 50.3 | 72.2 | 55.3 | 78.0 | 71.4 | 85.4 | 39.5 | 78.6 | 86.2 | 64.1 | 68.6 |

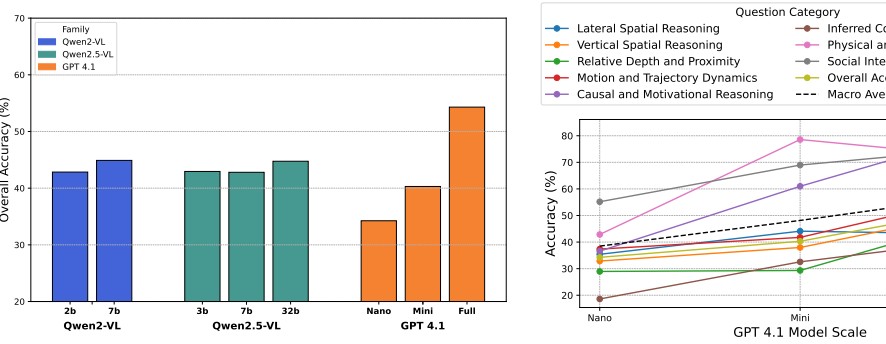

(a) Overall performance vs Model Scale  (b) Category-wise performance vs Model Scale

Figure 7: Impact of Model Scale.

**Human Baseline** Non-expert annotators benchmarked themselves against the `ImplicitQA` dataset using our `Implicit-Eval` tool. The annotators took around 1 minute per question on average, and achieved an overall score of 83%, with macro average of 85.6%. The best models lag behind human baseline the most in lateral spatial reasoning, relative depth and motion dynamics.

**Overall Performance** Table 3 highlights that the reasoning based GPT O3 outperformed all other models significantly, achieving an overall accuracy of 64.1% and a macro average of 68.6%. GPT 4.1 also exhibited robust performance, reaching overall accuracy of 54.3% followed by Gemini 2.5 Flash and Claude 4 sonnet. Open-source models generally performed worse than proprietary models. Early-generation of VideoLLMs like LLaVA-NeXT-Video perform especially poorly, achieving an overall score of 33.9%, indicating limited ability to reason beyond surface-level cues. Current generation open source VideoLLMs perform similarly on the Macro-Average metric (all within a range of 46.0 to 50.2), however they display variation across categories; Qwen, Gemma, Internvl models excel at spatial reasoning and viewpoint questions, while LLaVA-OneVision based models (LLaVA-Video is finetuned from LLaVA-OneVision) succeed at Motion, Motivational Reasoning.

Implicit reasoning remains an unsolved challenge even for leading VideoQA models. More frames or larger models alone are insufficient, we need architectural innovations or novel training paradigms. Despite being state-of-the-art in traditional VideoQA, no open-source model crosses 50% on `ImplicitQA`, reflecting the dataset's challenge and the limitations of current architectures in handling implicit, unstated, or cross-frame reasoning.

**Model Capacity vs. Implicit Reasoning** GPT models notably benefited from larger scales, with GPT 4.1 OpenAI (2024a) demonstrating substantial accuracy improvements, indicating more parameters facilitate effective implicit reasoning. Among open-source models, larger scales like Qwen2.5-VL Bai et al. (2025)-32b consistently outperformed smaller variants (7b, 3b) by a small margin.

**Category-wise Analysis** The performance across distinct implicit reasoning categories reveals varying levels of difficulty as shown in Figure 7 (b),Figure 8.

**(a) Vertical Spatial Reasoning**: Among open-weight models, LLaVA-OV Li et al. (2024a) performed best with 50.46% accuracy. In proprietary, GPT O3 excelled with a superior accuracy of 72.2%.

**(b) Relative Depth and Proximity**: GPT-based models demonstrated notable dominance in this category , particularly GPT O3 (55.3%), followed by GPT 4.1 OpenAI (2024a) (51.1%). Among open-weight models, Qwen2-VL achieved a commendable 40.6%, but lagged proprietary models.

**(c) Motivational Reasoning**: This was particularly challenging for all models except GPT variants. GPT O3 notably outperformed all others, achieving 85.4%, significantly ahead of the next best GPT 4.1 at 82.9%. Open-weight models achieved considerably lower performance.

**(d) Inferred Counting**: This category presented substantial difficulty across models, with GPT variants outperforming others yet still indicating room for improvement (maximum of 41.9% for GPT models). Open-weight models generally struggled more, highlighting a significant performance gap.

The experimental results underscore the inherent complexity and nuanced challenges associated with implicit reasoning in videos. They emphasize the clear advantage of proprietary GPT models, notably GPT O3, in handling complex implicit video reasoning tasks, attributed to their larger scale and deeper contextual understanding. Open-source models like Qwen2.5-VL, while competitive in certain categories, still show substantial room for improvement.

**Impact of Reasoning Prompt on GPT Models** We introduced a structured reasoning prompt specifically emphasizing spatial relationships and narrative summarization before answering questions. GPT 4.1-Mini OpenAI (2024a), with the added reasoning prompt, showed enhanced accuracy across multiple categories compared to the standard GPT 4.1-Mini. Specifically, improvements

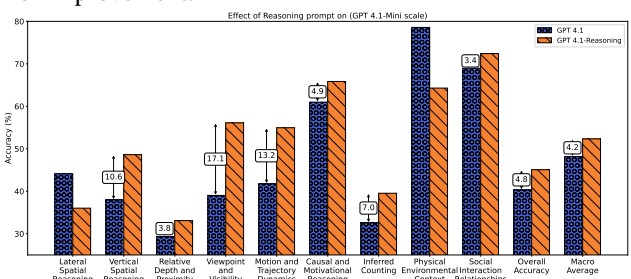

Figure 8: Effect of Reasoning Prompt

were evident in vertical spatial reasoning (from 37.96% to 48.61%), viewpoint and visibility (from 39.02% to 56.1%), and motion and trajectory dynamics (from 41.76% to 54.95%) as shown in Figure 8. This clearly indicates that structured prompting focused on spatial and narrative reasoning significantly enhances model performance. Please refer to Section D for detailed reasoning prompt.

**Effect of Number of Frames:** Going from 8 to 16 frames improves performance modestly. From 16 frames to 32 frames, accuracy saturates or slightly improves, but not significantly. For Qwen2.5 VL model it improves 2%.

**Uneven performance across reasoning categories** Models perform relatively better on Social Interaction and Motivational Reasoning, possibly reflecting pretraining biases towards human-centric scenarios. Categories involving numerical inference (Inferred Counting) and nuanced spatial reasoning (Relative Depth and Proximity) consistently show lower accuracy, highlighting a need for improvement.

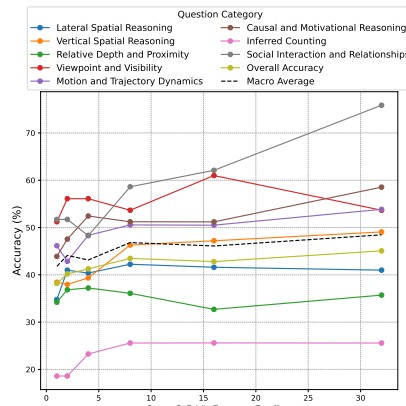

Figure 9: Effect of Frame Scaling

## 5 CONCLUSION

We introduce ImplicitQA, a novel benchmark for evaluating implicit reasoning in videos. Prior datasets largely target explicit visual understanding, leaving a gap in reasoning over narrative, cinematic content where answers often depend on inferred motivations and relationships. Across extensive experiments with contemporary models, we find that state-of-the-art systems struggle on implicit reasoning: scaling model size and extending temporal context offer only modest gains. Weaknesses are especially pronounced for categories requiring nuanced numerical inference, fine-grained spatial reasoning, and integration of long-term narrative context. ImplicitQA provides a rigorous evaluation protocol and baseline analyses to catalyze research that moves VideoQA beyond surface-level recognition toward deeper, human-like narrative understanding.

## REPRODUCIBILITY STATEMENT

Fully anonymized source code is provided in the supplementary material. We also include the `ImplicitQA` benchmark, with data curation process described in detail in Section 3. Code for all model evaluations is provided, and the evaluation protocols are detailed in Section 4, A ensuring that results can be easily reproduced. Upon acceptance, we will release the dataset, evaluation scripts, and annotation tools.

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

We organize the appendix as follows:

## A  DATA, TOOL AND LICENSES

We plan to publicly release the benchmark, annotation tool, evaluation scripts and evaluation tool with Apache 2.0 license upon acceptance. The employed multi-modal LMMs are released under Apache 2.0 license. We ran all our evaluations on NVIDIA A6000 48GB GPUs, and will release the eval scripts for reproducibility.

## B  DETAILED RESULTS

We present detailed results for varying model scales, temporal context across different models in Table 4. The best-performing open-source models were variants of the Qwen2.5 VLWang et al. (2024a) family at 32B scale. Model scale had a noticeable but diminishing effect: Within Qwen2.5 VL, increasing model size from 3B to 32B parameters provided modest performance gains (approximately +2% accuracy improvement). Smaller models, such as LLaVA-OneVision, struggled significantly in challenging reasoning categories irrespective of scale. Distinct performance variations across reasoning categories emerged. Social Interaction was relatively easier, with accuracy up to 79.31% (qwen2.5 VL).

Analysis on increasing frame count generally improves model performance for certain architectures, but this improvement plateaus or slightly degrades beyond a threshold. For instance, LLaVA-NeXT-VideoZhang et al. (2024a) exhibited a peak performance at 16 frames (33.9%) with a marginal decrease at 32 frames (32.56%). Similarly, LLaVA-OneVisionLi et al. (2024a) performed slightly better at 32 frames (43.16%) compared to 16 frames (43.4%), indicating negligible performance gains. For Qwen2.5-VLBai et al. (2025) 32B model, increasing frames from 4 to 32 resulted in accuracy improvements, particularly evident in the inferred counting & social interaction categories, achieving substantial accuracy improvements from 27.91% to 41.86% and from 62.07% to 79.31%, respectively. This suggests deeper frame context substantially aids in specific implicit reasoning tasks.

## C  IMPLICITQA DETAILED STATISTICS

This section provides additional statistics for `ImplicitQA` benchmark, highlighting the dataset's diversity across multiple dimensions, including

- Genre
- Media type
- Movie Release Timeline
- Difficulty based on hard-ness score

Table 4: Detailed Results on `ImplicitQA` for all implicit reasoning categories on various Video-oLMMs in multiple settings.

| Model | Scale | #Frames | Lateral Spatial Reasoning | Vertical Spatial Reasoning | Relative Depth and Proximity | Viewpoint and Visibility | Motion & Traj. Dynamics | Motivational Reasoning | Inferred Counting | Physical & Env. Context | Social Interaction & Relations | Avg. | Macro Avg. |
|---|---|---|---|---|---|---|---|---|---|---|---|---|---|
| LLaVA-NeXT-Video Zhang et al. (2024a) | 7b | 8 | 34.78 | 31.48 | 28.95 | 43.90 | 37.36 | 40.24 | 23.26 | 42.86 | 55.17 | 33.72 | 37.56 |
| | 7b | 16 | 36.00 | 29.60 | 30.10 | 48.80 | 36.30 | 39.00 | 30.20 | 35.70 | 51.70 | 33.90 | 37.50 |
| | 7b | 32 | 34.78 | 29.63 | 30.08 | 39.02 | 36.26 | 36.59 | 27.91 | 35.71 | 37.93 | 32.56 | 34.21 |
| LLaVA-OneVision Li et al. (2024a) | 7b | 16 | 37.30 | 46.80 | 35.00 | 56.10 | 57.10 | 57.30 | 23.30 | 50.00 | 55.20 | 43.40 | 46.40 |
| | 7b | 32 | 35.40 | 50.46 | 33.08 | 53.66 | 56.04 | 56.10 | 18.60 | 50.00 | 65.52 | 43.16 | 46.54 |
| LLaVA-Video Zhang et al. (2024b) | 7b | 8 | 31.68 | 41.20 | 29.32 | 48.78 | 57.14 | 57.32 | 13.95 | 50.00 | 62.07 | 39.02 | 43.50 |
| | 7b | 16 | 36.00 | 44.00 | 31.60 | 56.10 | 60.40 | 62.20 | 14.00 | 50.00 | 62.10 | 42.10 | 46.30 |
| | 7b | 32 | 36.02 | 47.22 | 32.71 | 51.22 | 58.24 | 63.41 | 16.28 | 57.14 | 68.97 | 43.27 | 47.91 |
| Qwen2.5-VL Bai et al. (2025) | 3b | 16 | 39.75 | 43.98 | 33.83 | 63.41 | 52.75 | 56.10 | 20.93 | 57.14 | 65.52 | 42.95 | 48.16 |
| | 7b | 1 | 34.78 | 38.43 | 34.21 | 51.22 | 46.15 | 43.90 | 18.60 | 57.14 | 51.72 | 38.18 | 41.80 |
| | 7b | 2 | 40.99 | 37.96 | 36.84 | 56.10 | 42.86 | 47.56 | 18.60 | 64.29 | 51.72 | 40.19 | 44.10 |
| | 7b | 4 | 40.37 | 39.35 | 37.22 | 56.10 | 48.35 | 52.44 | 23.26 | 42.86 | 48.28 | 41.25 | 43.14 |
| | 7b | 8 | 42.24 | 46.30 | 36.09 | 53.66 | 50.55 | 51.22 | 25.58 | 57.14 | 58.62 | 43.48 | 46.82 |
| | 7b | 16 | 41.60 | 47.20 | 32.70 | 61.00 | 50.50 | 51.20 | 25.60 | 42.90 | 62.10 | 42.80 | 46.10 |
| | 7b | 32 | 40.99 | 49.07 | 35.71 | 53.66 | 53.85 | 58.54 | 25.58 | 42.86 | 75.86 | **45.07** | 48.46 |
| | 32b | 4 | 39.75 | 44.91 | 39.10 | 48.78 | 41.76 | 57.32 | 27.91 | 57.14 | 62.07 | 43.27 | 46.53 |
| | 32b | 8 | 38.51 | 45.83 | 40.98 | 56.10 | 47.25 | 59.76 | 23.26 | 50.00 | 62.07 | 44.54 | 47.08 |
| | 32b | 16 | 38.51 | 48.15 | 37.59 | 51.22 | 49.45 | 62.20 | 32.56 | 50.00 | 62.07 | 44.75 | 47.97 |
| | 32b | 32 | 39.75 | 45.83 | 35.71 | 43.90 | 49.45 | 64.63 | 41.86 | 57.14 | 79.31 | 44.86 | **50.84** |
| Qwen2-VL Wang et al. (2024a) | 2b | 16 | 34.78 | 43.98 | 47.37 | 60.98 | 39.56 | 43.90 | 16.28 | 57.14 | 51.72 | 42.84 | 43.97 |
| | 7b | 1 | 38.51 | 39.35 | 36.84 | 53.66 | 42.86 | 48.78 | 13.95 | 42.86 | 55.17 | 39.66 | 41.33 |
| | 7b | 2 | 36.65 | 40.28 | 40.60 | 46.34 | 50.55 | 50.00 | 20.93 | 42.86 | 58.62 | 41.57 | 42.98 |
| | 7b | 4 | 39.13 | 45.83 | 39.47 | 46.34 | 49.45 | 51.22 | 20.93 | 50.00 | 58.62 | 43.05 | 44.56 |
| | 7b | 8 | 37.27 | 47.69 | 40.98 | 53.66 | 48.35 | 54.88 | 16.28 | 50.00 | 65.52 | 44.11 | 46.07 |
| | 7b | 16 | 39.80 | 46.80 | 40.60 | 51.20 | 52.70 | 58.50 | 16.30 | 35.70 | 72.40 | 44.90 | 46.00 |
| | 7b | 32 | 40.37 | 49.07 | 40.98 | 48.78 | 48.35 | 60.98 | 16.28 | 35.71 | 62.07 | 44.96 | 44.73 |
| **Proprietary Models** | | | | | | | | | | | | | |
| GPT 4.1 OpenAI (2024a) | Full | 16 | 42.90 | 53.20 | 51.10 | 48.80 | 59.30 | 82.90 | 41.90 | 71.40 | 75.90 | 54.30 | 58.60 |
| | Mini | 16 | 44.10 | 37.96 | 29.32 | 39.02 | 41.76 | 60.98 | 32.56 | 78.57 | 68.97 | 48.14 | 48.14 |
| | Nano | 16 | 35.40 | 32.87 | 28.95 | 58.54 | 37.36 | 36.59 | 18.60 | 42.86 | 55.17 | 34.25 | 38.48 |
| GPT O3 OpenAI (2024b) | Full | 16 | 50.30 | 72.20 | 55.30 | 78.00 | 71.40 | 85.40 | 39.50 | 78.60 | 86.20 | **64.10** | **68.60** |

## C.1 GENRES

To further characterize the content diversity in `ImplicitQA`, we manually annotate both the primary and secondary genres of each video. We assign these genre annotations by considering genres listed on dedicated pages for each movie on publicly available sources such as IMDb[1] and Wikipedia[2].

We assign each video a primary genre, which represents a movie's core theme and structure; and a secondary genre which reflects additional aspects of a movie. We observe that these primary and secondary genres come from a total of 15 different genres which are listed(in alphabetical order) as follows:

- Action
- Adventure
- Black comedy
- Comedy
- Crime
- Drama
- Fantasy
- Horror
- Mystery
- Psychological horror/thriller
- Romance
- Sci-fi
- Socio-political
- Thriller
- Western

Figure 10 shows the primary and secondary genre distribution of our dataset. We observe that the dataset includes a wide range of genres, with Comedy, Action, and Adventure being the most prominent primary, while Comedy, Adventure and Fantasy being the top 3 secondary. This broad

---

[1]https://www.imdb.com/

[2]https://www.wikipedia.org/

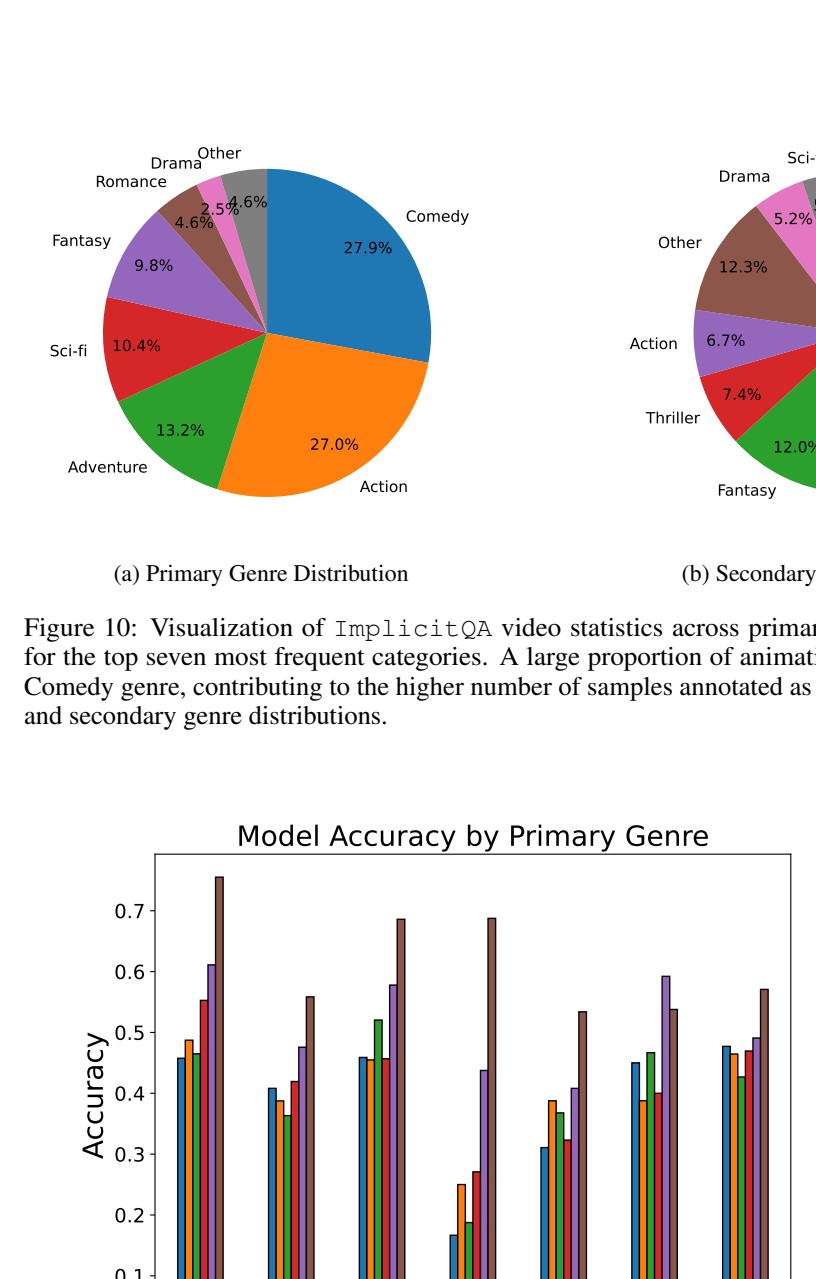

(a) Primary Genre Distribution

(b) Secondary Genre Distribution

Figure 10: Visualization of `ImplicitQA` video statistics across primary and secondary genres for the top seven most frequent categories. A large proportion of animation videos fall under the Comedy genre, contributing to the higher number of samples annotated as Comedy in both primary and secondary genre distributions.

Figure 11: Model accuracy across primary video genres in the `ImplicitQA` dataset. Performance varies significantly by genre, with O3 model consistently leading across genres except Romance.

genre coverage ensures the benchmark captures diverse narrative structures, thematic elements, and stylistic conventions - essential for evaluating implicit reasoning across contexts. To investigate how genre influences model performance, we show accuracy across primary genre categories in `ImplicitQA`. As shown in Figure 11, genre plays a substantial role in performance variation. Overall models perform best on Action, Comedy, and Romance. In contrast, performance drops for genres like Drama and Fantasy. Notably, the O3 model outperforms all others across every genre except Romance, suggesting its stronger ability to generalize across narrative structures. The variation across genres also underscores the importance of content diversity in benchmark design, as genre-specific reasoning challenges reveal gaps in current video LMMs capabilities.

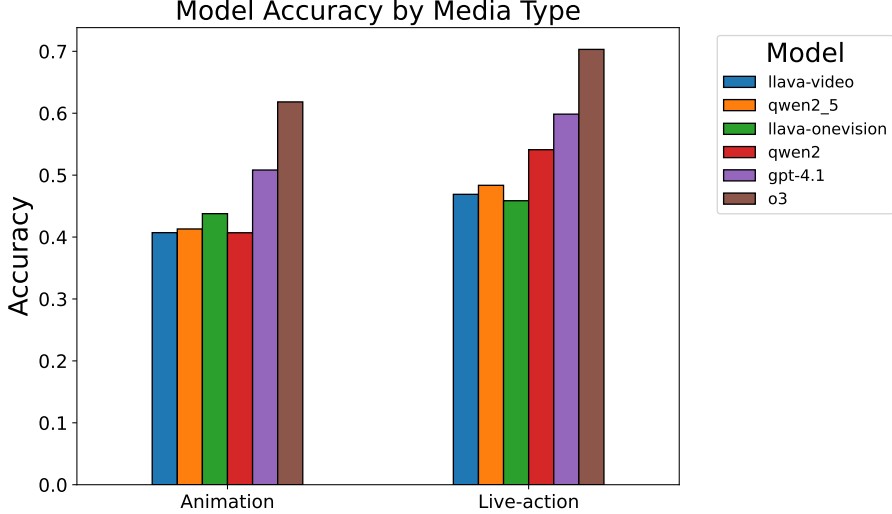

Figure 13: Model accuracy across media types (Animation vs. Live-action). Performance is consistently higher on live-action videos, with the largest gains observed in higher-capacity models such as GPT-4.1 and O3.

## C.2 MEDIA TYPE

We further categorize the videos into Live-Action and Animation to highlight the diversity in visual domains present in `ImplicitQA`. As shown in Figure 12, the dataset maintains a balanced composition across both categories, with 58.3% of the videos being animated and 41.7% live-action. This mix ensures exposure to varied stylistic, motion, and rendering characteristics that challenges LMMs.

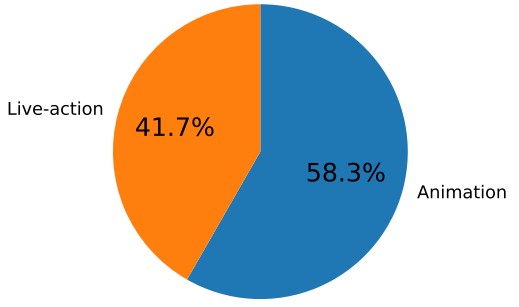

Figure 12: Distribution of Media Type in `ImplicitQA`.

To further understand model generalization across different visual domains, we show performance on animation and live-action videos. As shown in Figure 13, all models demonstrate stronger performance on live-action content. The gap is especially more for larger models like GPT-4.1 and O3, which outperform others by a substantial margin. These results indicate that models may rely more effectively on grounded visual signals and realistic spatial cues present in live-action videos, whereas stylized representations in animation pose additional challenges for implicit reasoning. This highlights the need for further adaptation for animation-rich inputs.

## C.3 MOVIE RELEASE TIMELINE

We annotate the release year for each video and present the distribution by decade in Figure 14. The `ImplicitQA` dataset spans a broad temporal range, covering films from the 1960s to current decade. A film's release period is often indicative of its visual and narrative style - including factors such as picture quality, cinematographic techniques, editing conventions, character costumes, and action design. This temporal diversity in `ImplicitQA` enhances its realism and ensures broader generalization by exposing models to varied cinematic styles and storytelling conventions across eras.

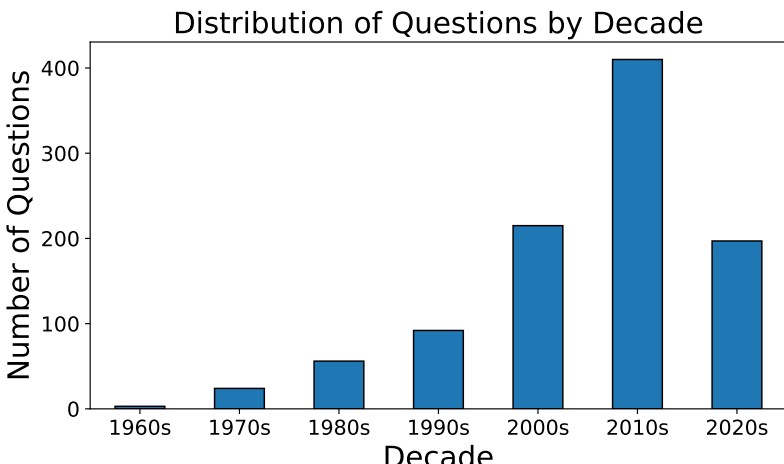

Figure 14: Distribution of videos in `ImplicitQA` by release year. The dataset spans over **_7 decades_**, capturing a wide range of visual styles, production techniques, and narrative conventions across different time periods.

## C.4 DIFFICULTY

To better understand model behavior under varying levels of difficulty, we propose a hardness-based partitioning of the `ImplicitQA` dataset. Each question is assigned a hardness score derived from model performance: questions answered incorrectly by all models contribute more to the score, while those answered correctly by all models contribute none. Specifically, the hardness score is computed by summing each incorrect model's average accuracy. This metric reflects how broadly difficult a question is across model architectures.

As shown in Figure 15, the distribution of hardness scores is approximately uniform across the three difficulty categories - Easy, Medium, and Hard - with roughly equal numbers of questions in each group. This balanced partitioning allows for a fair evaluation of model performance across difficulty levels.

Figure 16 shows model accuracy broken down by these categories. While all models perform well on Easy questions, accuracy drops substantially for Medium and Hard examples. GPT-4.1 and GPT-O3 demonstrate better generalization across difficulty levels, whereas other models perform near or below random chance on the hardest questions. These findings reveal a steep difficulty gradient and highlight the value of hardness-aware analysis for assessing reasoning robustness. To contextualize model performance, we compare GPT-O3's accuracy against human performance. As shown in Figure 17, GPT-O3 performs comparably to humans on Easy questions and shows only a modest drop on Medium questions. However, the gap be-

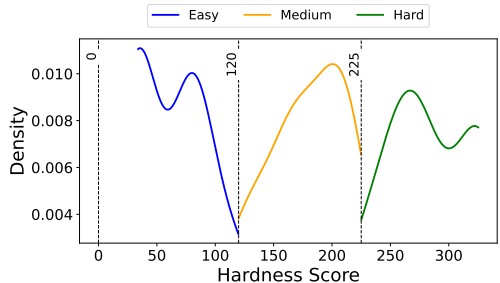

Figure 15: Density distribution of questions in `ImplicitQA` based on hardness scores. Questions are categorized into three difficulty levels - Easy (0–120), Medium (120–225), and Hard (225+) - based on model performance scores. The distribution is approximately uniform, ensuring a balanced evaluation across varying difficulty levels.

comes pronounced on Hard examples: while human accuracy remains relatively high, GPT-O3's performance declines sharply, approaching random chance. It is important to note that the human baseline reflects responses from non-expert participants, while ground truth annotations in `ImplicitQA` were created by expert annotators with domain familiarity. The relatively strong performance of

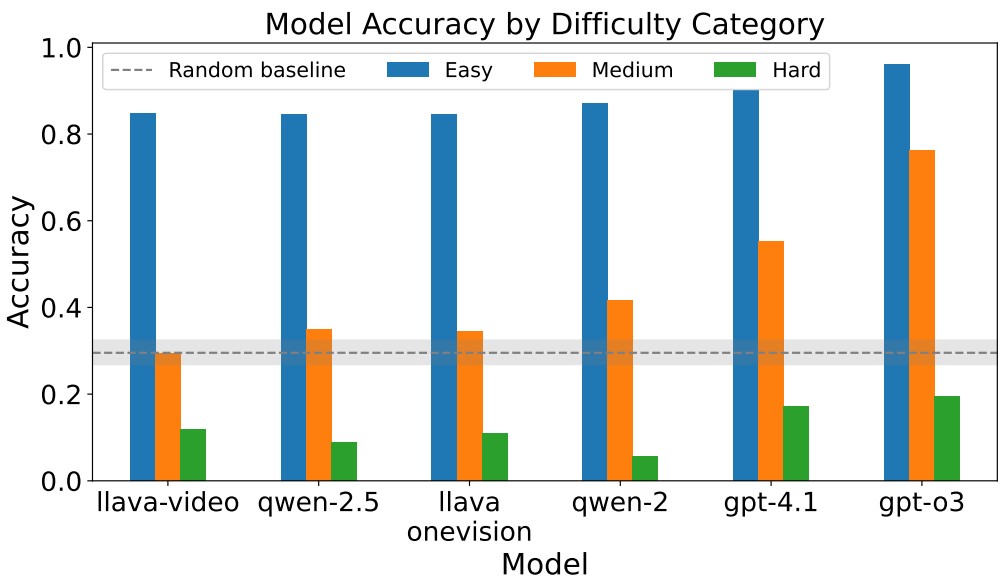

Figure 16: Model accuracy across difficulty categories. While all models perform strongly on Easy questions, performance drops significantly on Medium and Hard examples.

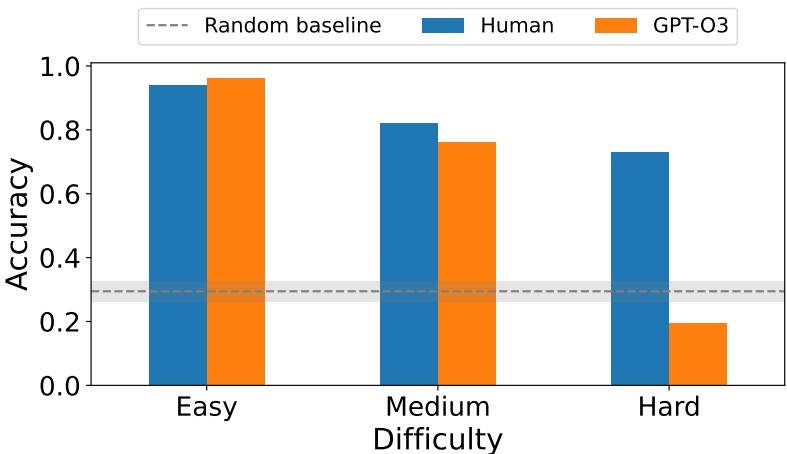

Figure 17: Accuracy comparison between GPT-O3 and non-expert human annotator across difficulty categories. While both perform well on Easy and Medium questions, human accuracy remains robust on Hard questions, whereas GPT-O3 performance drops significantly. Ground truth annotations were provided by expert annotators, underscoring the reasoning gap between models and even non-expert humans on complex questions.

non-experts highlights the accessibility of implicit reasoning for humans, even without expertize, while also emphasizing the performance ceiling that current models have yet to reach.

## C.5 QUESTION WORD DISTRIBUTION

In Figure 18 we present the word count distribution for questions in `ImplicitQA`.

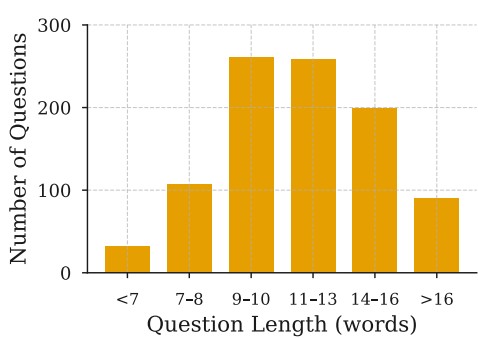

Figure 18: `ImplicitQA` Question word counts

## D    IMPACT OF REASONING PROMPT

The reasoning prompt used to evaluate the impact of structured reasoning on GPT models is illustrated in Figure 19. This prompt is specifically designed to guide the model in analyzing video content for inferred reasoning across the various categories defined in `ImplicitQA`. It breaks down the task into sequential steps-analyzing the video, summarizing key spatial relationships, highlighting important elements, answering the multiple choice question, and providing additional insights-thereby encouraging systematic focus on implicit reasoning.

The output format enforces a structured response, including a concise summary, bullet-pointed key themes and spatial cues, the selected answer, and contextual reasoning. This structured approach is intended to enhance the model's ability to infer unstated relationships and improve its overall accuracy.

As demonstrated in Figure 8 of the main paper, incorporating this reasoning-based prompt yields significant performance gains, improving the accuracy of GPT-4.1 Full by 3.9% and GPT-4.1 Mini by 4.8%. A detailed breakdown of the results is provided in Table 5. In Figure 20, we present a qualitative example illustrating the effectiveness of the reasoning prompt. When prompted without reasoning, GPT-4.1 incorrectly identifies the spatial relationship between characters as perpendicular. With the structured reasoning prompt, the model successfully breaks down spatial positions, directionality, and frame context to arrive at the correct answer: They are facing directly toward each other. This example highlights how the reasoning prompt not only improves accuracy but also fosters interpretability by making the model's decision-making process more transparent and spatially grounded.

Table 5: Results with Reasoning Prompt on `ImplicitQA` for all implicit reasoning categories.

| Model | Scale | Lateral Spatial Reasoning | Vertical Spatial Reasoning | Relative Depth and Proximity | Viewpoint and Visibility | Motion & Traj. Dynamics | Motivational Reasoning | Inferred Counting | Physical & Env. Context | Social Interaction & Relations | Avg. | Macro Avg. |
|---|---|---|---|---|---|---|---|---|---|---|---|---|
| GPT 4.1 OpenAI (2024a) | Full | 42.90 | 53.20 | 51.10 | 48.80 | 59.30 | 82.90 | 41.90 | 71.40 | 75.90 | 54.30 | 58.60 |
| GPT 4.1-Reasoning OpenAI (2024a) | Full | 50.90 | 63.00 | 51.10 | 46.30 | 63.70 | 79.30 | 41.90 | 71.40 | 86.20 | **58.20** | **61.50** |
| GPT 4.1 OpenAI (2024a) | Mini | 44.10 | 37.96 | 29.32 | 39.02 | 41.76 | 60.98 | 32.56 | 78.57 | 68.97 | 40.30 | 48.14 |
| GPT 4.1-Reasoning OpenAI (2024a) | Mini | 36.02 | 48.61 | 33.08 | 56.10 | 54.95 | 65.85 | 39.53 | 64.29 | 72.41 | **45.07** | **52.32** |

## E    EXPERIMENT STATISTICAL SIGNIFICANCE

In Table 6, we report the statistical error margins for the Qwen-2.5 VL model. To assess the variability of the model's performance, we conducted five independent runs using the same evaluation setup. For each of the nine implicit reasoning categories in `ImplicitQA` as well as the overall accuracy, we compute the mean and standard deviation.

## F    HUMAN BASELINE

To establish a reference point for model performance, we evaluated with non-expert human annotator on the `ImplicitQA` benchmark using our custom-built Implicit-Eval tool. The annotator was not

```
system_prompt = """
Assist users in understanding and gaining insights from video content, with
particular emphasis on inferring the relative spatial positioning of various
characters, objects, etc., and answer related multiple choice questions.

Guide users in digesting and analyzing the content of video material by
breaking down the key themes, summarizing the narrative or subject matter,
identifying important elements such as spatial relationships, characters,
plot points, or pivotal moments, and answering any multiple choice questions
related to the video.

# Steps

1. Analyze Video Content: Identify main themes, characters, and narrative
structure with a focus on spatial relationships.
2. Summarize: Write a brief summary of the video's key points and overall
narrative, emphasizing spatial positioning.
3. Highlight Key Elements: Point out significant moments or elements in the
video, particularly those that reveal spatial positioning and relationships.
4. Answer Multiple Choice Question: Review the provided multiple choice
question and select the most accurate answer, considering spatial
inferences.
5. Provide Insights: Offer any additional insights or context, specifically
regarding spatial relationships, that could help in comprehending the
video's content.

# Output Format

Provide a structured text output that includes:
- A brief summary paragraph of the video focusing on spatial positioning.
- Bullet points highlighting key themes, spatial relationships, characters,
and significant elements.
- The selected answer for the multiple choice question.
- Additional spatial relationship-focused insights as a paragraph.

# Notes

- Ensure summaries are concise but comprehensive, with an emphasis on
spatial understanding.
- Focus on elements that are crucial to understanding the video's spatial
intent or message.
- Use clear and precise language to define spatial positioning.
- Make sure the multiple choice answer is clearly indicated and explained
through spatial reasoning.
"""
```

Figure 19: Reasoning prompt used to guide GPT models in analyzing video content. The prompt breaks down the task into structured steps - focusing on spatial relationships, narrative, summarizing key elements, selecting the correct answer, and providing insights - encouraging systematic reasoning aligned with `ImplicitQA`'s implicit question categories.

provided with prior exposure to the dataset or answer keys and completed the evaluation at a natural pace, averaging approximately one minute per question. The annotator achieved an overall accuracy of 83.0%, with a macro-average score of 85.6% across reasoning categories. While performance was consistently strong across most categories, Inferred Counting emerged as the most challenging for the human. This aligns with the category's reliance on temporal cues and implicit aggregation of visual information across multiple scenes, factors that often test not only reasoning but also memory and attention.

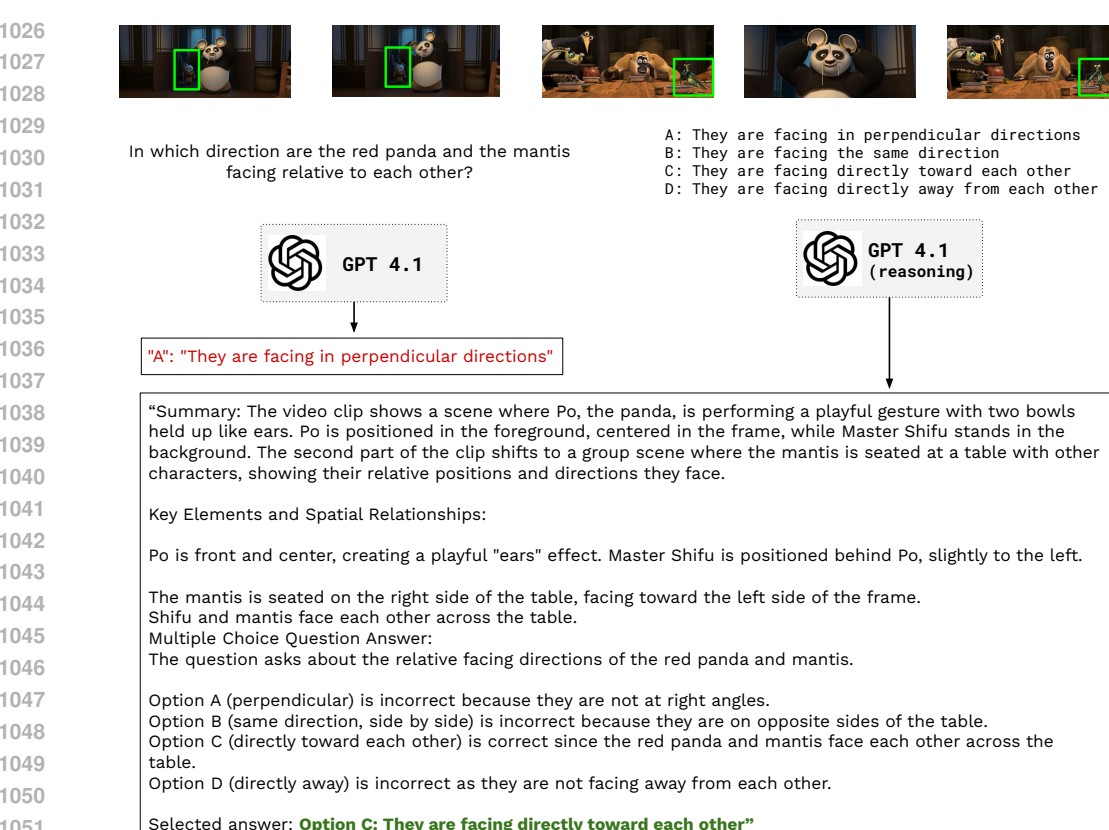

Figure 20: Qualitative example from `ImplicitQA` demonstrating how the reasoning prompt improves GPT-4.1's performance. Without structured reasoning, the model incorrectly selects Option A. With the reasoning prompt, the model provides a detailed spatial analysis of character positions and orientations, ultimately selecting the correct answer, Option C. This showcases the benefit of guiding the model through spatially grounded reasoning steps.

Table 6: Mean and Standard Deviation Across Categories (in %)

|  | Lateral Spatial Reasoning | Vertical Spatial Reasoning | Relative Depth and Proximity | Viewpoint and Visibility | Motion & Traj. Dynamics | Motivational Reasoning | Inferred Counting | Physical & Env. Context | Social Interaction & Relations | Avg. |
|---|---|---|---|---|---|---|---|---|---|---|
| Mean | 41.61 | 47.41 | 32.71 | 60.00 | 50.55 | 51.71 | 26.51 | 42.86 | 62.07 | 42.93 |
| Std Dev | 0.00 | 0.25 | 0.00 | 1.34 | 0.00 | 0.67 | 1.27 | 0.00 | 0.00 | 0.12 |

We further analyze the human accuracy across questions grouped by difficulty level using our hardness-based scoring method. As shown in Figure 17, the non-expert human achieved near-ceiling performance on Easy questions and maintained strong performance on Medium ones. Remarkably, even on Hard questions defined by consistent failure across models, the human annotator performed well above chance, achieving over 70% accuracy. This analysis underscores the significant gap between human and model capabilities on complex reasoning tasks. Even non-expert humans demonstrate strong generalization and implicit understanding, particularly in scenes that demand spatial, temporal, or motivational reasoning, highlighting the limitations of current state-of-the-art Video LMMs.

These results indicate that, while challenging, the questions are well-defined, solvable, and carefully validated requiring real reasoning capabilities.

## G    ANNOTATION TOOL INTERFACE

As discussed in Section 3.1 in the Main paper, we have designed and built an annotation tool for intuitive and efficient workflow, which allows annotators to follow a structured approach towards formulating multiple choice QA pairs for desired video clip segments. The tool is optimized for fast navigation and efficient verification. Said annotation can be systematically performed by adhering to the following procedure. We have divided the procedure into 4 intuitive sub-procedures which further comprise of 3 steps each. We describe these subprocedures below.

1. **Video Download:** Our tool allows the user to input a video URL and download the associated video. The user needs to run backend.py, navigate to the web interface using the generated link, input desired URL and click on the "Download Video" button. The tool downloads the video and shows it in the embedded display.
2. **Clip Segment Selection:** The user can put time markers at any desired place using the provided time bar, and viewing the video in the embedded display. The user can pause, rewind and analyze the video frame-by-frame to determine the best possible clip segment suiting to their requirements. Then the user can preview their selected clip segment by clicking the "Preview Selection" button.
3. **QA Annotation:** After selecting the desired segment, the user can formulate the appropriate question and type it in the designated space. The tool has provision to input as many choices in the designated spaces as needed, using the "Add Answer Choice" button. After finalizing, the user can click the "Save Annotation button to record their multiple choice QA pair.
4. **Verification:** The generated QA pair can be viewed in the mini-display at the bottom. Once the user has recorded all their annotations for the video clip, they can verify said annotations by clicking on the "View Annotations" Tab at the top of the page. There, they can see all the videos annotated in that particular session. Clicking on the "View Annotations" button associated to a video takes the user to the page for that video where all annotations are listed along with the embedded display which can play the relevant clip. Clicking on a question displays all details associated with that question including the relevant timestamps ensuring complete verification.

## H    QUALITATIVE RESULTS

To illustrate the complexity and diversity of implicit reasoning required in our benchmark, we present qualitative examples spanning all nine reasoning categories in Figure 22,23,24. For each example, we show the relevant video frames, question, answer choices, correct answer, and model predictions. These examples highlight a wide range of reasoning challenges -from spatial positioning and motion inference to social understanding and inherent explanation. Notably, GPT-O3 demonstrates superior performance in most cases.

### H.1    VIEWPOINT AND VISIBILITY

In the Viewpoint and Visibility example shown in Figure 22, only GPT-O3 correctly infers the adopted perspective of the panda character, showcasing its ability to track camera shifts and narrative cues.

### H.2    PHYSICAL AND ENVIRONMENTAL CONTEXT

In the Physical and Environmental Context scenario as in Figure 22, GPT-O3 again outperforms others by correctly identifying the white car driven by the woman in black, leveraging spatial cues across frames.

### H.3    VERTICAL SPATIAL REASONING

As shown in Figure 22, all models successfully answer a Vertical Spatial Reasoning question involving relative positions in a multi-level scene.

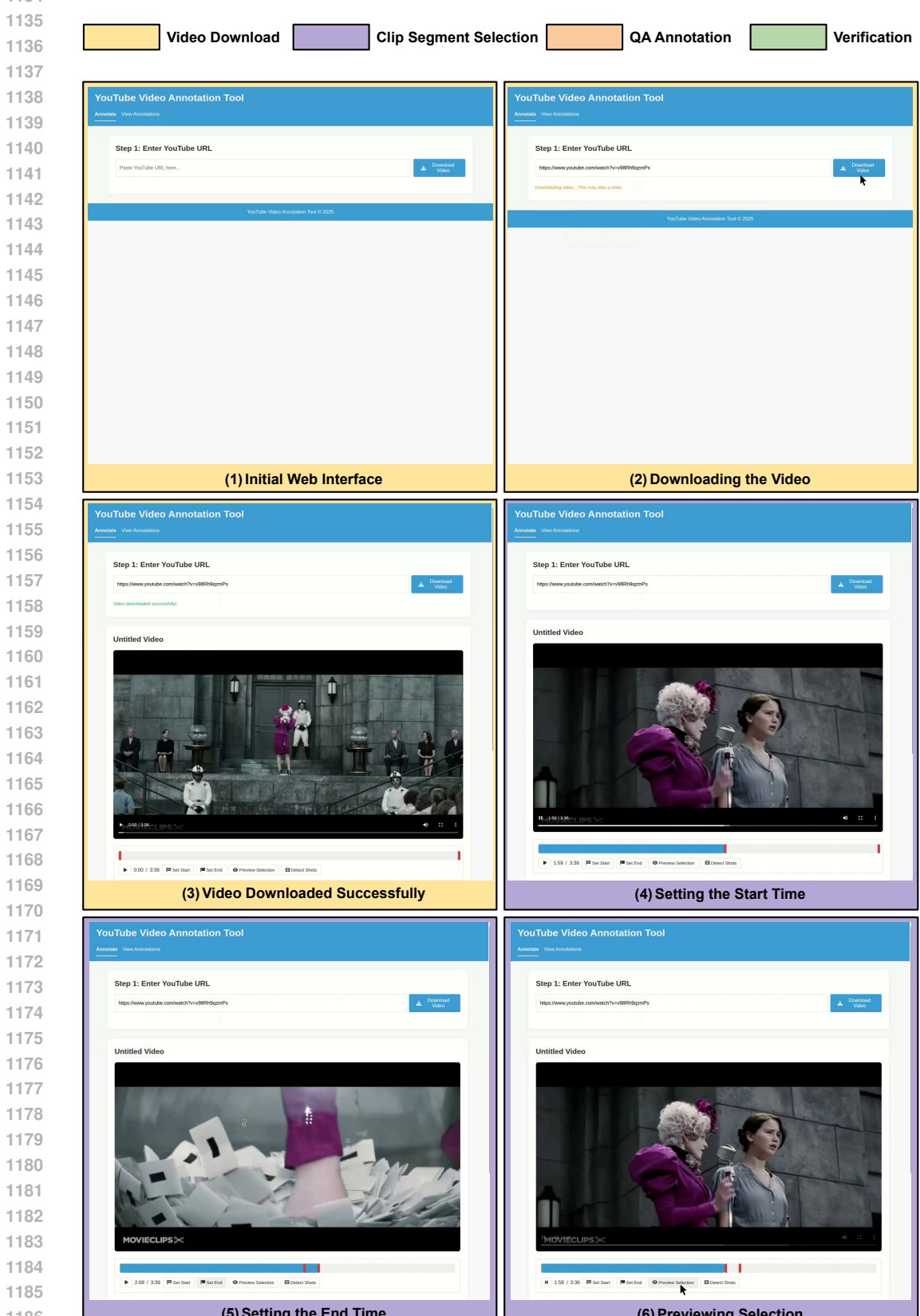

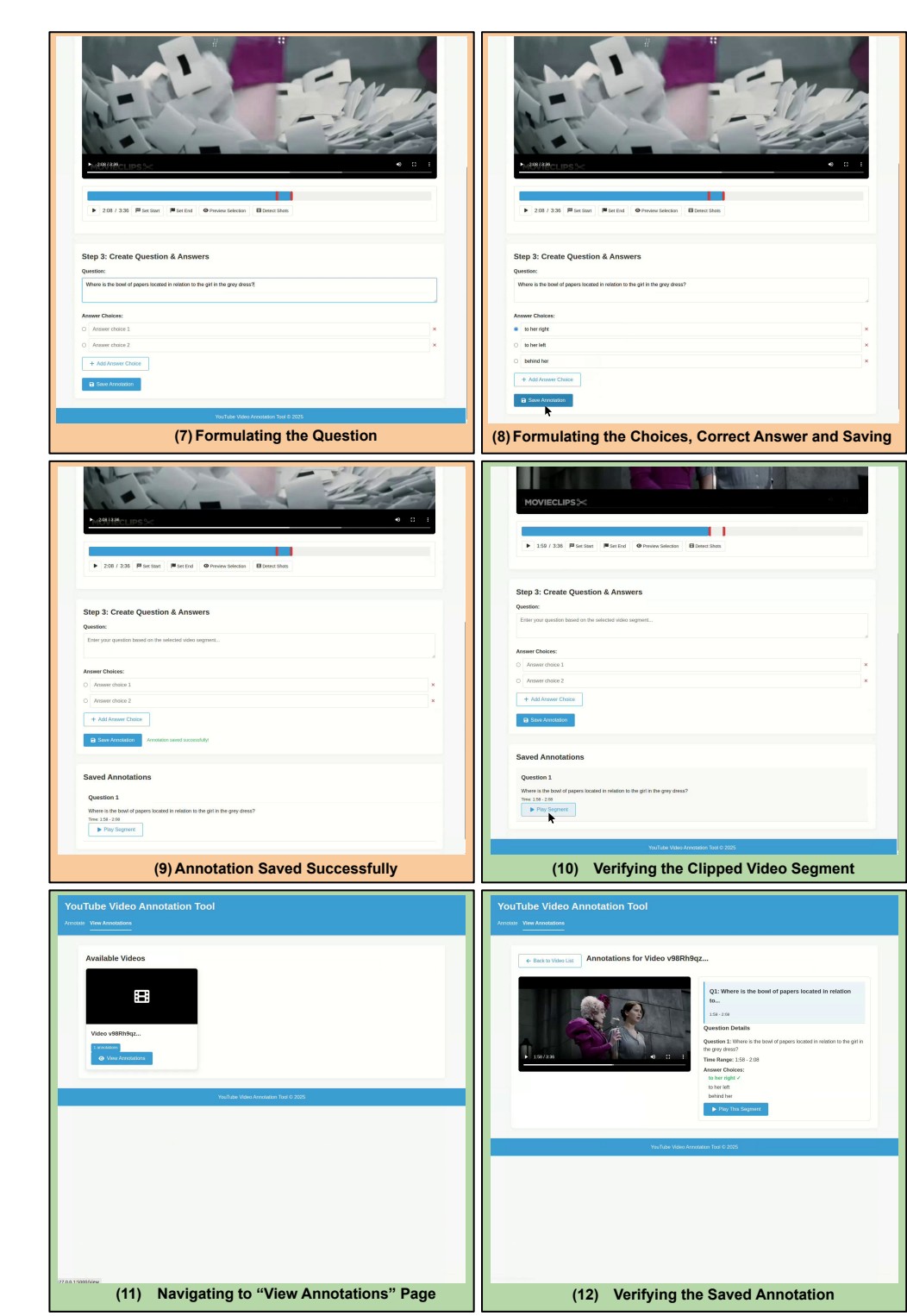

Figure 21: **Schematic illustration of the annotation workflow using our FrameQuiz tool.** The process is organized into **four sub-procedures**: Video Download, Clip Segment Selection, QA Annotation, and Verification. Each sub-procedure contains three steps, resulting in a **12-step pipeline** for generating high-quality multiple-choice QA pairs from video clips. Final annotations are stored locally following verification.

### H.4    RELATIVE DEPTH AND PROXIMITY

In contrast, more nuanced categories reveal sharper contrasts in performance as shown in Figure 23. For Relative Depth and Proximity, GPT demonstrates strong spatial inference by accurately localizing characters and interpreting their orientations.

### H.5    LATERAL SPATIAL REASONING

For Lateral Spatial Reasoning as shown in Figure 23, we see that most model get it correct except GPT-4.1.

### H.6    MOTION AND TRAJECTORY DYNAMICS

In the Motion and Trajectory Dynamics example shown in Figure 23, most models correctly track the direction of movement, though GPT-O3 misjudges the path—suggesting sensitivity to camera motion.

### H.7    SOCIAL INTERACTION AND RELATIONSHIPS

The Social Interaction and Relationships case shown in Figure 24, involving subtle facial cues and body language, is correctly answered only by GPT-O3 and GPT-4.1, reflecting their advanced multimodal understanding.

### H.8    INFERRED COUNTING

For Inferred Counting shown in Figure 24, models struggle to aggregate information across frames, with GPT-O3 and Qwen-VL identifying the correct number, while GPT-4.1 undercounts.

### H.9    MOTIVATIONAL REASONING

Finally, in the Motivational Reasoning example in Figure 24, GPT-O3 and GPT-4.1 correctly attribute the escape of the rats to being discovered, while others fail to connect with the relevant event. These examples collectively highlight the diversity and difficulty of implicit reasoning tasks in `ImplicitQA`.

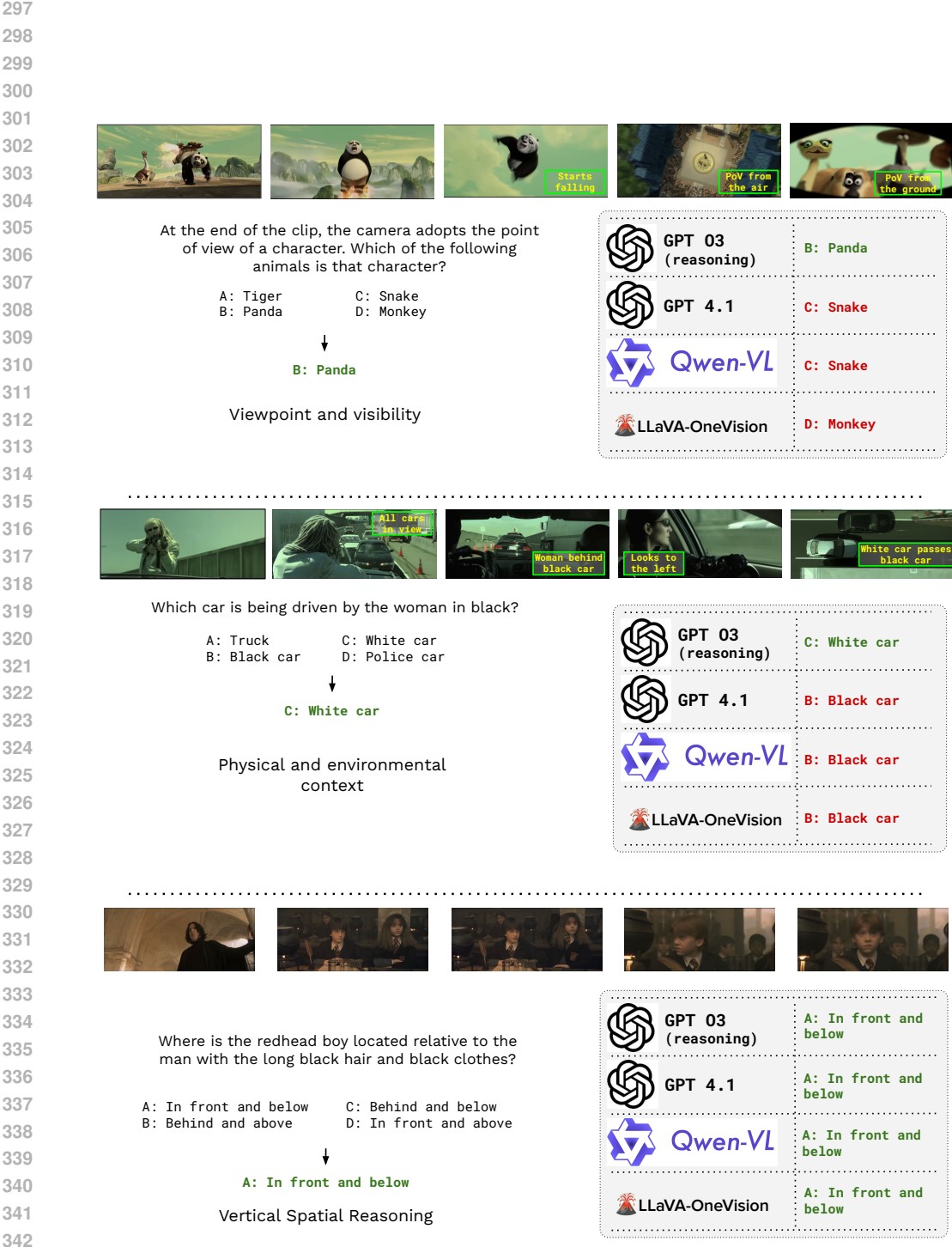

Figure 22: More Qualitative `ImplicitQA` examples, targeting distinct implicit-reasoning dimensions.

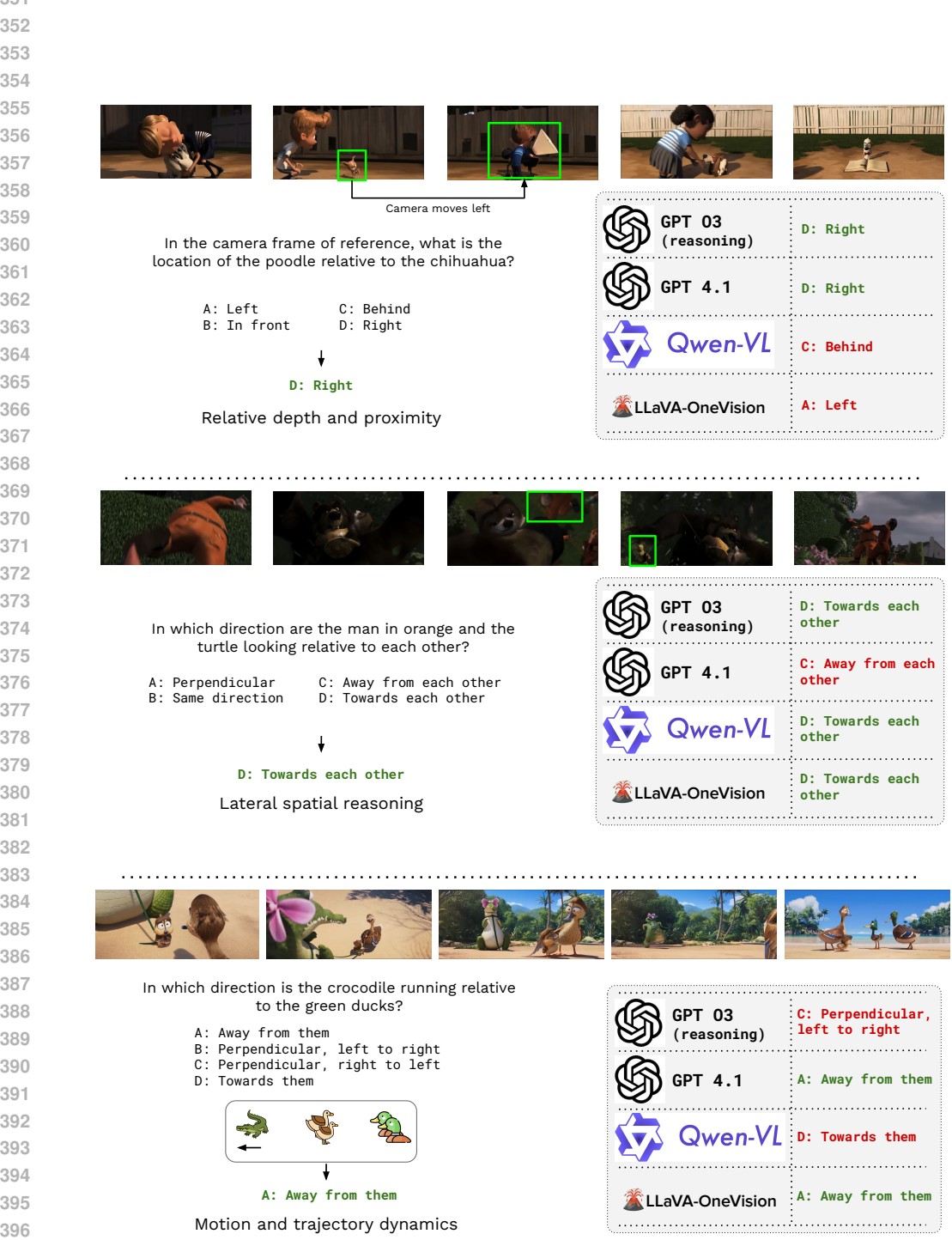

Figure 23: More Qualitative ImplicitQA examples, targeting distinct implicit-reasoning dimensions.

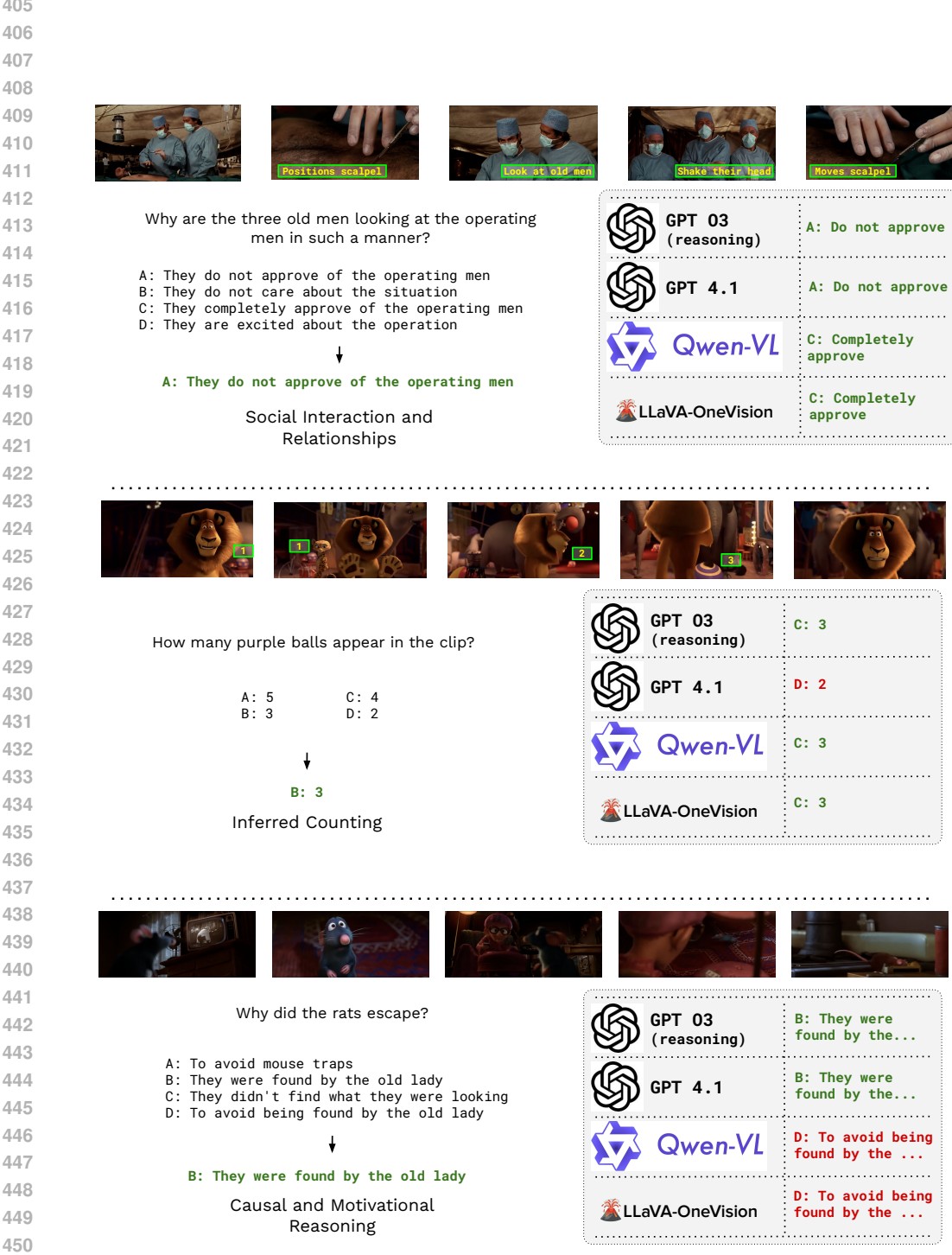

Figure 24: More Qualitative `ImplicitQA` examples, targeting distinct implicit-reasoning dimensions.

