# OpenReview forum: "ImplicitQA: Going beyond frames towards Implicit Video Reasoning"
_ICLR.cc/2026/Conference — ICLR 2026 Conference Withdrawn Submission_

### Official Review · Reviewer_Y1QG · 2025-10-21

**Soundness:** 3
**Presentation:** 3
**Contribution:** 2
**Rating:** 2
**Confidence:** 5

**Summary:**

This paper presents ImplicitQA, a benchmark designed to evaluate Video Question Answering (VideoQA) models on implicit reasoning within narrative video content, such as movies. Unlike previous benchmarks that focus on explicit, surface-level visual cues, ImplicitQA challenges models to infer information such as motivational reasoning, spatial relationships, and inferred counting that is suggested but not directly depicted across discontinuous frames. The authors manually curated 1,000 question-answer pairs across 15 genres and 7 decades of film to create the testbed. Evaluations on numerous state-of-the-art VLMs show a significant performance gap compared to human baselines, with no open-source model exceeding 50% accuracy, highlighting the limitations of current architectures in achieving deeper narrative understanding.

**Strengths:**

- New dataset focusing on implicit reasoning
- The authors meticulously and manually annotate 1000 QAs
- The dataset is quite diverse with 15 genres across 7 decades of content
- The benchmarking is robust with 7 open-source models and 4 proprietary ones plus human-level performance
- The paper is well-written and easy to follow

**Weaknesses:**

- I think novelty is an issue here as several datasets have already explored the claims made by the authors: CRIPP-VQA [1] and PhysBench [2] explore physics-based properties and directions covering Lateral and Vertical Reasoning, Depth, Viewpoint, Trajectory, Inferred counting and Physical and Environmental Context. Additionally, VCR [3] is a large-scale dataset that covers Social Interaction and Relationships as well as motivational reasoning, among other interesting implicit QA types.
- I understand the challenge of manually annotating videos, yet I feel the dataset size is too small and tries to cover too much for its size. It would have been of more value to the community if the dataset had focused on a niche topic, then the size would not have been an issue to me.
- Stemming from the previous weakness, about 50% of the data is about two classes (Fig. 5). The dataset aims to be diverse but does not really achieve that with the less represented class having 10 times less QAs than the most represented one.
- The abstract claims 1K meticulously annotated QA pairs drawn from 1K high-quality creative video clips. In Figure 3 (c), I see that some videos have as many as 17 questions. Does that mean some videos do not have any QAs? If so, why?
- A lot of QA are (arguably) direct (not implicit). For instance: How many individuals are in the scene?, Where did the group of girls go?...
- [Minor] An ethics statement would be nice, since the authors employ humans for annotation and use videos from the internet.
- [Minor] Line 912: Expertize (instead of expertise)

[1] Patel, Maitreya, et al. "Cripp-vqa: Counterfactual reasoning about implicit physical properties via video question answering.", EMNLP’22

[2] Chow, Wei, et al. "Physbench: Benchmarking and enhancing vision-language models for physical world understanding.", ICLR’25

[3] Zellers, Rowan, et al. "From recognition to cognition: Visual commonsense reasoning.", CVPR’19

**Questions:**

Weaknesses +
- Could the authors highlight the differences between ImplicitQA and the existing benchmarks that focus on implicit video QAs (see weakness #1)?
- Why would vision-only be considered as a feature? Would having subtitles or audio alter the results? If so, the community would benefit from having these options.

---

### Official Review · Reviewer_Srmt · 2025-10-29

**Soundness:** 2
**Presentation:** 1
**Contribution:** 2
**Rating:** 2
**Confidence:** 3

**Summary:**

This paper introduces ImplicitQA, a video question answering benchmark specifically targeting implicit visual reasoning—i.e., questions that cannot be answered by cues visible in a single frame or short clip. The dataset contains 1,000 multiple-choice QA pairs from 1,000 movie clips, annotated by experts and organized into nine reasoning categories (e.g., lateral/vertical spatial reasoning, relative depth, motion/trajectory, viewpoint/visibility, causal/motivational, social interactions, physical context, inferred counting). Extensive evaluations show a large human–model gap: human baseline ≈83%, while the best model (GPT-o3) reaches ~64% overall accuracy, and scaling frames/parameters yields only modest gains. The authors plan to release the dataset and tooling.

**Strengths:**

Originality. First benchmark laser-focused on implicit, cross-frame visual inference with a clear taxonomy and visual-only constraint (no audio/subtitles), which isolates vision-centric reasoning.
Quality. Careful expert curation and validation; an end-to-end pipeline for authoring questions and distractors; broad, category-wise analysis across many models; sensible probes on reasoning prompts and frame scaling.
Clarity. The paper clearly states contributions, shows motivating examples, dataset stats, and a transparent curation workflow.
Significance. Reveals a substantial human–model gap on implicit reasoning and demonstrates that simply adding frames/parameters is insufficient, likely spurring architectural/training innovations.

**Weaknesses:**

Scale & headroom. At 1k items, the dataset is relatively small for today’s regime; conclusions about scaling might change with a larger, more diverse corpus. Consider expanding to strengthen statistical power and allow train/val/test splits for learning-on-ImplicitQA.
Multiple-choice format. The benchmark uses MCQ with author-crafted distractors, which can introduce guessing artifacts and distractor bias; adding open-ended or rationale tasks would reduce format shortcuts.
Domain coverage. Clips come from movies; cinematic conventions differ from other long-video domains (egocentric, sports, instructional). A cross-domain extension would clarify generalization.
Operational definition of “implicit.” The notion is central yet difficult to formalize. The paper mentions expert relabeling, but more IAA (inter-annotator agreement) and rubric details would bolster reliability.
Modal ablations. The visual-only choice is principled, but many cinematic inferences hinge on dialogue/sound design. A controlled ablation (visual-only vs. +audio/+subs) would quantify the incremental difficulty.
Baselines & training. Results are mostly zero-shot evaluations. It would be informative to train (or adapt) at least one open model on a train split to establish a learnability baseline and prevent over-interpreting zero-shot gaps.

**Questions:**

1.	Definition & QA policy. What exact rubric distinguishes implicit from explicit items (with examples that were excluded)? Do you report IAA for this labeling decision?
2.	Distractor quality. How did you validate distractors to avoid annotation artifacts or unintended textual shortcuts (e.g., option length/frequency)? Any adversarial filtering beyond expert design?
3.	Sampling bias. What is the distribution across genres/decades and how does accuracy vary by them? Would you consider adding non-movie domains to assess transfer?
4.	Modal ablations. Since the benchmark is visual-only, can you report a small-scale ablation adding subtitles/audio to quantify the gap attributable to non-visual cues?
5.	Learning curves. If you create a public train split, how do models improve with supervised fine-tuning or preference/RL signals? Do reasoning prompts remain beneficial once models are adapted?
6.	Temporal sensitivity. How sensitive are results to frame sampling rate/stride and to clip length per category (e.g., counting vs. spatial)? Any per-category scaling curves?
7.	Licensing & release. For movie clips, what usage and redistribution policy will accompany the public release (links + timestamps vs. video snippets)?

---

### Official Review · Reviewer_HMjT · 2025-10-29

**Soundness:** 3
**Presentation:** 3
**Contribution:** 4
**Rating:** 8
**Confidence:** 3

**Summary:**

The work introduces a video dataset as a benchmark aimed evaluating implicit reasoning in Video Question Answering task.
The authors collect 1000 annotated QA pairs from 1000 creative video clips across 15 genres and 7 decades
Questions are categorized into  testing one of nine reasoning dimensions: lateral and vertical spatial reasoning, depth and proximity, viewpoint and visibility, motion and trajectory, causal and motivational reasoning, social interactions, physical context, and inferred counting.
Evaluations of 11 models (7 open and 4 closed source) reveal consistent under-performance even top models like GPT-O3 reach only 64.1% average accuracy, while human baseline achieve 83.0%.

While the introduce benchmark is small in size, with a clear motivation and high quality annotation, it can provide substantial diagnostic insights to pinpoints where and why current systems fail at human-like reasoning. With the strong capabilities of current VideoQA model, i can see the value this benchmark could provide by adding a new dimension to the problem.

**Strengths:**

The gap between explicit and implicit reasoning in video understanding is an interesting problem to tackle especially with the growing capabilities of modern systems. The focus on narrative inference and off-screen causality is novel and impactful.

Comprehensive evaluation across 30 model configurations including number of frames, impact of reasoning prompts mitigates performance degradation of models from other sources and lends credibility and relevance to the datasets proposed purpose of implicit reasoning.

Diverse dataset that includes animation and live-action, temporal and stylistic diversity strengthens the dataset's generalizability.

**Weaknesses:**

It is not abundantly clear on if there were any inter-annotator agreements on answer correctness and relevance of video segments. It would have been helpful if there was a quantitative agreement scores.

Limited theoretical framing on how the nine distinct reasoning categories were decided. While each category is well defined within the paper, the work would benefit from connecting implicit reasoning to broader theories in cognitive science or narrative comprehension currently the framing is largely empirical.

Although high quality, 1K samples are modest for a modern benchmark. The limited size could potentially restrict robust statistical comparisons but as stated in the summary this does not significantly affect my rating.

**Questions:**

How did you define “implicit” to the annotators and how to ensure the definition was closely followed, was there a checklist or rule-based criterion during annotation?

What were the number of expert and non-expert annotators?

Why are Lateral Spatial Reasoning and Vertical Spatial Reasoning considered separately, what is the distinction in the challenge provided between the two? Intuitively, I see no difference in the between the two and fall under the same umbrella of spatial reasoning.

Were the distractor options validated for plausibility?

Do all questions require multi-frame reasoning, or can some be inferred within a single frame?

---

### Official Review · Reviewer_V3sQ · 2025-11-01

**Soundness:** 2
**Presentation:** 3
**Contribution:** 2
**Rating:** 2
**Confidence:** 4

**Summary:**

This paper presents ImplicitQA, a benchmark designed to test human-like implicit reasoning where viewers are required to infer spatial information, causal relationships, etc. from multiple video frames where the answer to the question is not explicitly shown in a single frame. The authors conduct experiments to show performance differences between human baseline and models, and additional studies examining (1) impact of reasoning prompt on GPT models, (2) effect of number of frames, and (3) uneven performance across reasoning categories.

**Strengths:**

1. This paper is clearly written and easy to follow.
2. The authors and annotators manually curated 1k QA pairs across 1k videos, and defined 9 reasoning categories to ensure diversity in the QA types.
3. The authors highlight the performance gap between human and SOTA models demonstrating current models’ deficiency in reasoning over cinematic content where answers could only be inferred from across disjoint visual contexts.

**Weaknesses:**

1. One of my main concerns is to what extent the performance gap reflects insufficient “implicit reasoning” as opposed to differences in how models allocate visual attention, such as focusing sparsely versus in detail.\
    (i). In the 6 examples shown in the paper, I personally was unable to answer correctly (a) Lateral Spatial Reasoning from Figure 1, and (a) Physical and Environment Context from Figure 6. This difficulty appears partially due to the high level of visual attention to detail required from viewers. No additional analysis has examined the effect of this factor. (See more in Question 1)\
    (ii). The authors have also observed “Uneven performance across reasoning categories” and reported relatively better performance on Social Interaction and Motivational Reasoning where attention and perceptual granularity are less detail-oriented.
2. Only one human baseline is provided. As the non-expert annotators were only tested with the video setting but not frame setting like what models receive, it is unclear how much of the performance gap between humans and models arises from their differences in processing visual inputs vs. limitations in “implicit reasoning”. (See more in Question 3)
3. ImplicitQA is largely composed of spatial reasoning QAs according to Figure 5. Since the authors have not provided qualitative or quantitative definitions of “implicit reasoning”, it remains unclear at the moment how this work could be differentiated from the vast amount of literature on spatial/-temporal reasoning benchmarks such as [1], [2].
4. The claims that “no open-source model exceeds 50% accuracy” and “Open-source models generally performed worse than proprietary models” are not sufficiently supported by experiments when open-source model results reported in the main table are all 7B and the biggest open-source model tested in the paper being 32B Qwen2.5-VL. Considering [3] has reported slightly better results achieved by Qwen2-VL-72B compared to GPT-4o on temporal reasoning, the authors would need more experiments to make such claims.\
\
[1] Wu, Bo, et al. "Star: A benchmark for situated reasoning in real-world videos." arXiv preprint arXiv:2405.09711 (2024).\
[2] Mao, Jiayuan, et al. "Clevrer-humans: Describing physical and causal events the human way." Advances in Neural Information Processing Systems 35 (2022): 7755-7768.\
[3] Shangguan, Ziyao, et al. "Tomato: Assessing visual temporal reasoning capabilities in multimodal foundation models." arXiv preprint arXiv:2410.23266 (2024).

**Questions:**

1. Following up on Weakness 1(i)., have the authors considered conducting experiments where captions are first generated for each frame, and models then reason only in a “text-only” setting to see if the details required to answer questions (such as (a) Lateral Spatial Reasoning from Figure 1, and (a) Physical and Environment Context from Figure 6) are already left out of caption generation step. This two-pass experiment could help clarify the source of the poor performance: sparse attention or “implicit reasoning” limitations.
2. The 6 example videos all appear to be animated videos. What is the percentage of natural videos compared to animated videos presented in this benchmark? Have the authors considered analyzing the effect of such features on model performances?
3. Have the authors considered including more baselines such as where non-expert annotators are only given the same frames as the models receive?

**Details Of Ethics Concerns:**

The benchmark includes clips from well-known copyrighted works such as Toy Story, Super Mario, Kung Fu Panda, etc., whose studios such as Disney/Pixar, Nintendo/Universal, and DreamWorks Animation are known to be very strict about video redistribution, even for academic use. However, the paper does not address any licensing, copyright, or redistribution compliance considerations associated with these materials.

---

### Note · Authors · 2025-11-14

I have read and agree with the venue's withdrawal policy on behalf of myself and my co-authors.